# A dentate gyrus-CA3 inhibitory circuit promotes evolution of hippocampal-cortical ensembles during memory consolidation

Hannah Twarkowski[1,2,3], Victor Steininger[1,2,3], Min Jae Kim[1,2,3], Amar Sahay[1,2,3,4]*

[1]Center for Regenerative Medicine, Massachusetts General Hospital, Boston, United States; [2]Harvard Stem Cell Institute, Cambridge, United States; [3]Department of Psychiatry, Massachusetts General Hospital, Harvard Medical School, Boston, United States; [4]BROAD Institute of Harvard and MIT, Cambridge, United States

**Abstract** Memories encoded in the dentate gyrus (DG) – CA3 circuit of the hippocampus are routed from CA1 to anterior cingulate cortex (ACC) for consolidation. Although CA1 parvalbumin inhibitory neurons (PV INs) orchestrate hippocampal-cortical communication, we know less about CA3 PV INs or DG – CA3 principal neuron – IN circuit mechanisms that contribute to evolution of hippocampal-cortical ensembles during memory consolidation. Using viral genetics to selectively mimic and boost an endogenous learning-dependent circuit mechanism, DG cell recruitment of CA3 PV INs and feed-forward inhibition (FFI) in CA3, in combination with longitudinal *in vivo* calcium imaging, we demonstrate that FFI facilitates formation and maintenance of context-associated neuronal ensembles in CA1. Increasing FFI in DG – CA3 promoted context specificity of neuronal ensembles in ACC over time and enhanced long-term contextual fear memory. *In vivo* LFP recordings in mice with increased FFI in DG – CA3 identified enhanced CA1 sharp-wave ripple – ACC spindle coupling as a potential network mechanism facilitating memory consolidation. Our findings illuminate how FFI in DG – CA3 dictates evolution of ensemble properties in CA1 and ACC during memory consolidation and suggest a teacher-like function for hippocampal CA1 in stabilization and re-organization of cortical representations.

*For correspondence:
asahay@mgh.harvard.edu

## Editor's evaluation

This paper will be of interest to scientists across systems neuroscience or to those interested in how one component of a neural circuit contributes to downstream functions longitudinally. The techniques used in this paper allowed the authors to characterize how increasing feed forward inhibition in the dentate gyrus-CA3 hippocampal circuit impacts the formation and maintenance of context-specific ensembles in CA1 and the anterior cingulate cortex without directly stimulating the circuit.

## Introduction

The hippocampus plays a critical role in the formation of new episodic memories by generating conjunctive representations of experiences and transferring these representations to neocortical sites for memory storage or consolidation (*Frankland and Bontempi, 2005*; *McClelland et al., 1995*; *Peyrache et al., 2009*; *Squire, 2004*; *Wilson and McNaughton, 1994*). Highly processed sensory information underlying experiences are encoded in the dentate gyrus (DG) – CA3 circuit as distinct or updated representations of prior experiences and subsequently, routed out of CA1

to prefrontal cortical sites such as the anterior cingulate cortex (ACC) for consolidation. Following learning, prefrontal cortical neuronal ensembles are thought to undergo time-dependent refinement and re-organization as memories transition from recent to remote configurations and become more generalized (*DeNardo et al., 2019*; *Frankland and Bontempi, 2005*; *Frankland et al., 2004*; *Maviel et al., 2004*; *McClelland et al., 1995*; *Morrissey et al., 2017*; *Takehara-Nishiuchi and McNaughton, 2008*; *Wiltgen et al., 2010*; *Winocur et al., 2010*). The hippocampus is theorized to play a continuous, potentially instructive role in transformation of memories from conjunctive, context-rich representations to semantic, gist-like structures (*Goode et al., 2020*; *Nadel and Moscovitch, 1997*; *Teyler and DiScenna, 1986*; *Tonegawa et al., 2018*; *Winocur et al., 2010*; *Winocur et al., 2007*). Indeed, a growing body of experimental evidence not only supports a sustained role for the hippocampus in remote memory retrieval (*Goshen et al., 2011*; *Vetere et al., 2021*) but also in dictating the extent to which remote memories retain contextual details of the original experience (*Guo et al., 2018*; *Koolschijn et al., 2019*; *Tompary and Davachi, 2017*).

Investigations into neural circuit mechanisms mediating memory consolidation has identified key roles for principal cells and inhibitory neurons in mediating hippocampal-cortical communication. Specifically, genetic silencing of engram-bearing dentate granule neurons post training was shown to impair the maturation of context-specific neuronal ensembles in prefrontal cortex (*Kitamura et al., 2017*). Chronic genetic inhibition of CA3 outputs impaired high-frequency network oscillations such as sharp-wave ripples, experience-related firing patterns in CA1 and long-term contextual fear memory (*Nakashiba et al., 2009*). Parvalbumin inhibitory neurons (PV INs) in CA1 are the most extensively studied with regard to their roles in memory consolidation. CA1 PV INs have been shown to orchestrate synchronous activation of principal neurons, promote sharp-wave ripples (SWR), and facilitate hippocampal-cortical communication through increased sharp-wave ripple – spindle coupling in CA1 – ACC networks (*Buzsáki, 2015*; *Çaliskan et al., 2016*; *Gan et al., 2016*; *Ognjanovski et al., 2017*; *Stark et al., 2014*; *Xia et al., 2017*). Chemogenetic inhibition of PV INs in CA1 or in ACC following training inhibited consolidation of contextual fear memory (*Ognjanovski et al., 2017*; *Xia et al., 2017*). Together, these observations begin to highlight the contribution of principal neurons and INs in CA1 and ACC in memory consolidation. However, we know significantly much less about the contributions of principal neurons – IN circuits in DG – CA3, let alone how PV INs in CA3 govern the evolution of neuronal ensembles in downstream CA1 – ACC networks over time.

Dentate granule cell recruitment of feed-forward inhibition (FFI) onto CA3 is temporarily enhanced following learning and is one inhibitory circuit mechanism linked to memory consolidation (*Guo et al., 2018*; *Ruediger et al., 2012*; *Ruediger et al., 2011*). In previous work, we found that viral mediated selective enhancement of dentate granule cell recruitment of PV INs in CA3 increased inhibition onto CA3 and enhanced strength and precision of remote memories (*Guo et al., 2018*). Using genetic ensemble-tagging approaches and immediate-early gene expression, we found that increasing FFI in DG – CA3 conferred context-specific reactivation of neuronal ensembles in hippocampal – ACC – basolateral amygdala networks at remote timepoints. Furthermore, we found that maintenance of engram-bearing cell connectivity with PV INs prevented the time-dependent decay of the hippocampal engram thereby suggesting a link between maintenance of the hippocampal memory trace and remote memory precision (*Guo et al., 2018*). As with new findings, we were motivated to investigate how FFI in DG – CA3 impacts ongoing neuronal activity, maintenance and specificity of neuronal ensembles in hippocampal-cortical networks during memory consolidation. In this study, we sought to address this goal by combining viral genetics to selectively enhance FFI in DG – CA3 and longitudinal *in vivo* 1 photon calcium imaging and *in vivo* electrophysiology to record ensemble properties in CA1 and ACC over time. We found that enhancing FFI facilitates the formation and maintenance of context-associated neuronal ensembles in CA1. Stable and precise CA1 ensembles, in turn, promoted acquisition of context-specific neuronal ensembles in ACC at remote timepoint and enhanced long-term contextual fear memory. Furthermore, using simultaneous LFP recordings we found enhanced ripple-spindle coupling thus demonstrating a direct role for DG – CA3 FFI in enhancing hippocampal-cortical communication. Our findings illuminate how FFI in DG – CA3 dictates evolution of ensemble properties in CA1 and ACC during memory consolidation and suggest a teacher-like function for hippocampal CA1 in stabilization and re-organization of cortical representations.

## Results

### Viral enhancement of FFI in DG − CA3 increases PV IN perisomatic contacts onto CA3 principal neurons

To begin to understand how FFI in DG − CA3 contributes to evolution of ensemble properties during memory consolidation, we leveraged our recent discovery of a learning-regulated molecular brake of dentate granule cell connectivity with stratum lucidum PV INs in CA3. Viral downregulation of the cytoskeletal protein, Ablim3, in the DG (using validated shRNA vs. non-target scrambled NT) resulted in increased dentate granule cell mossy fiber contacts with PV INs and an increase in PV IN puncta or contacts with CA3. Consistently, *ex vivo* electrophysiological recordings showed that these anatomical changes in FFI connectivity resulted in increased feed-forward inhibition onto CA3 (*Guo et al., 2018*) (schematized in *Figure 1a*). We injected mice with lenti-shNT or lenti-shRNA in the DG and quantified PV puncta, an anatomical marker for PV-mediated inhibition, in CA3, CA1 and ACC 2 weeks past injection. As previously shown (*Guo et al., 2018*), we found an increase in PV puncta in CA3 in shRNA injected mice [Two-tailed Mann-Whitney: p = 0.029, n = 4]. Consistent with prior reports (*Soltesz and Losonczy, 2018*), we found a significantly higher number of PV puncta in the deeper layer of CA1 compared to the superficial layer [Two-way repeated measures ANOVA, layer x virus effect: N.S.: main virus effect N.S., main layer effect p < 0.0001, n = 4]. Our manipulation did not affect PV puncta in CA1 or ACC [ACC: Two-tailed Mann-Whitney test: N.S., n = 4] (*Figure 1b*). Together, these data demonstrate that our viral manipulation increases FFI onto CA3 without affecting PV IN − principal neuron anatomical contacts in CA1 or ACC.

### Increasing FFI in DG − CA3 promotes foot-shock associated neuronal responses in CA1 and ACC during memory encoding

To investigate how enhancing FFI in DG − CA3 prior to learning influences CA1 and ACC ensemble activity during acquisition and recall of recent and remote memories, we performed calcium imaging in a cohort of mice in a contextual fear conditioning paradigm (*Figure 1d–e*). We delivered lentiviruses shRNA (shRNA) or non-target shRNA (shNT) into the DG in combination with viral expression of the genetically encoded calcium indicator GCaMP6f in CA1 or ACC (*Figure 1c*). Two weeks following bilateral viral injections in DG (lenti-shNT or control mice vs. shRNA or experimental mice) and unilateral injections into CA1 or ACC (AAV1.CamKII.GCaMP6f), we recorded neuronal activity (calcium transients) using miniaturized endoscopes in CA1 or ACC (see example *Videos 1–4*). We performed recordings at same day baseline (BL), during contextual fear conditioning in the training context, context A, and during recall in context A and a distinct, neutral context (context C) at recent (day 1) and remote timepoints (day 16).

On day 0, control and experimental groups of mice were trained to associate context A with foot-shocks. Behavioral analysis of freezing levels revealed that both groups of mice responded similarly to foot-shocks in the training context A (*Figure 2a and c*). Consistent with prior work (*Jimenez et al., 2020*; *Pan et al., 2020*), our recordings of calcium transients during contextual fear conditioning revealed a subset of neurons in both CA1 and ACC that exhibited significant elevation in activity in response to foot-shocks (Foot-shock responsive neurons or FSR neurons) in both groups of mice [Two-tailed paired t-test, ACC shNT preFS vs postFS: p = 0.004, n = 5, ACC shRNA preFS vs postFS: p = 0.013, n = 5, CA1 shNT preFS vs postFS: p = 0.007, n = 5, CA1 shRNA preFS vs postFS: p = 0.024, n = 5](*Figure 2b, d and e*). However, the fraction of FSR neurons in ACC was significantly increased in the shRNA group compared to shNT group [Two-tailed unpaired t-test with Welch's correction, p = 0.043, n = 5] (*Figure 2f*, *Figure 2—figure supplement 1a*). Further analysis of FSR neurons revealed that the increase in FSR neuron activity in CA1 and ACC was accompanied by an increase in co-activity of these neurons i.e. increase in number of FSR correlated pairs [Two-tailed paired t-test, ACC shNT: p = 0.021, n = 5, ACC shRNA: N.S., n = 5, CA1 shNT: p = 0.021, n = 5, CA1 shRNA: p = 0.001, n = 5] (*Figure 2g–h* and *Figure 2—figure supplement 1b*). Note, this was not seen in ACC for shRNA group presumably because of high activity levels prior to foot-shocks. Shuffled data indicate that the increase in correlated pairs was mostly due to increased number of calcium events upon foot-shock. Additionally, the shRNA mice exhibited a significantly higher increase in correlated neurons in CA1 mice [Two-tailed unpaired t-test with Welch's correction, p = 0.044, n = 5] (*Figure 2—figure supplement 1b*). We did not detect increased co-activity of FSR with nonFSR neurons in CA1 or ACC

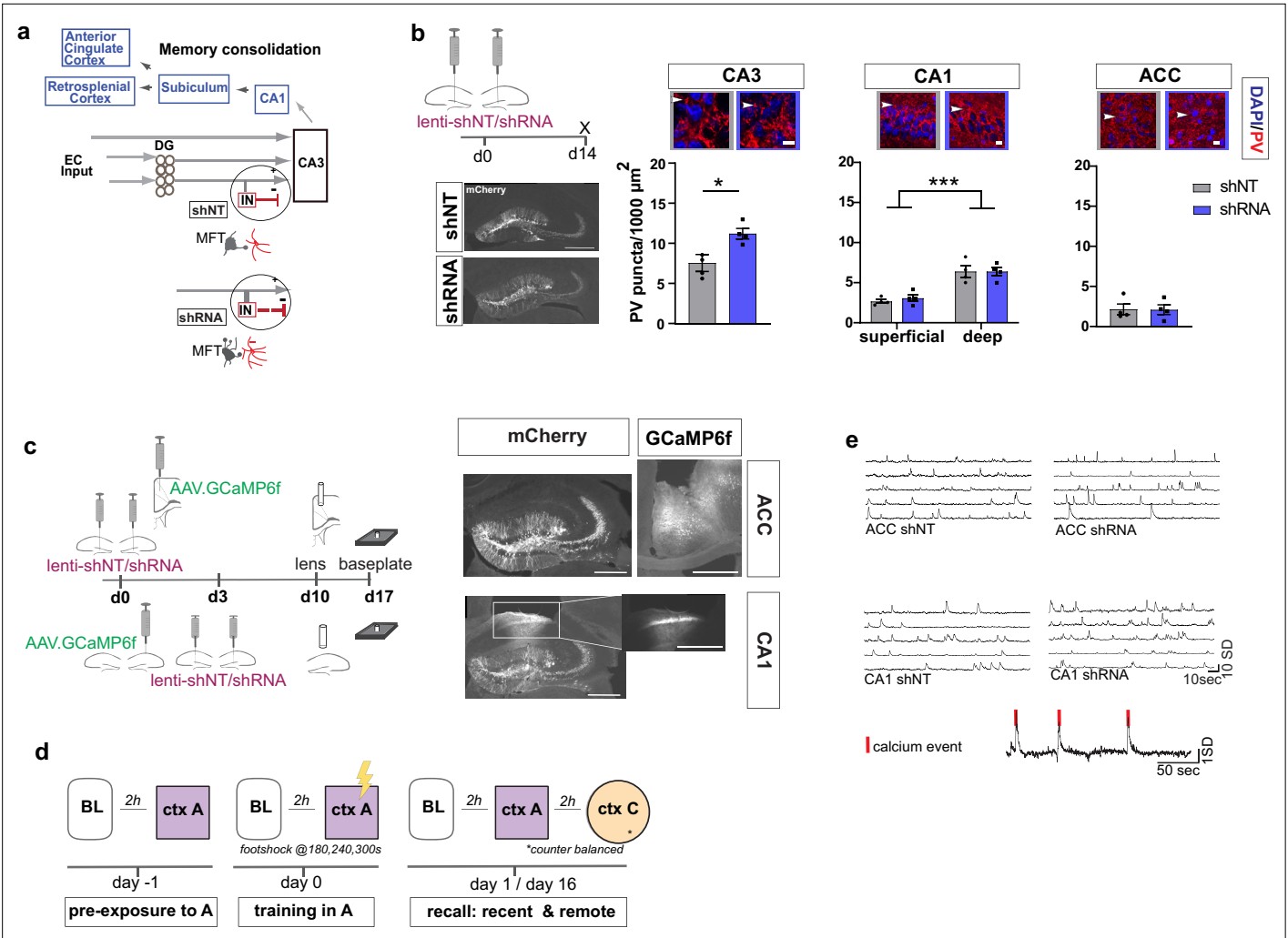

**Figure 1.** Anatomical characterization of PV IN contacts and *in vivo* calcium imaging platform for recording neural activity. (**a**) Simplified wiring diagram of hippocampal-cortical circuits supporting memory consolidation. Highlighted is the DG-PV IN-CA3 inhibitory microcircuit. Dentate granule cell (DGC) projections (Mossy fibers, MF) synapse onto pyramidal neurons in CA3 and also release glutamate directly onto stratum lucidum PV-positive interneurons (PV INs, red) through filopodia that emanate from mossy fiber terminals (MFTs, gray). Injection of lenti-shRNA into dentate gyrus (DG) increased MFT filopodia onto PV INs. In response to increased DGC excitatory drive PV INs elaborate inhibitory synaptic contacts in CA3 and feed-forward inhibition onto CA3 neurons (feed-forward inhibition, FFI). Pyramidal neurons in CA3 project to CA1 that sends projections to multiple brain regions such as the anterior cingulate cortex (ACC). (**b**) Quantification and representative images of PV puncta (white arrows) in CA3 (left), the deeper and superficial layer of CA1 (middle) and ACC (right) per ROI (1000 μm², scale bar: 10 μm) in behaviorally naïve mice 2 weeks post injection (left panel, scale bar: 500 μm). Images show representative examples of ROIs from shNT (left) and shRNA (right) injected mice. Arrows indicate PV puncta example. Bar graphs represent mean ± SEM. Scatter represent individual mice. Injection of lenti-shRNA significantly increased PV puncta in CA3 [Two-tailed Mann-Whitney: p = 0.029, n = 4] but not in CA1 [Two-way repeated measures ANOVA, layer x virus effect p = 0.583: virus main effect p = 0.752, layer main effect: p < 0.0001]; or ACC [Two-tailed Mann-Whitney test: p = 0.886, n = 4]. In CA1, a significant difference was found between layers. (**c**) Schematic workflow and example of virus injection and lens implantation for calcium imaging. Mice received unilateral injection of AAV1.CaMKII. GCaMP6f.WPRES.SV40 (GCaMP6f, left panel) in the region of interest and bilateral injection of lenti-shNT or lenti-shRNA virus (mCherry, right panel) in the hilus of dorsal DG. In CA1 implanted mice, the injections were separated by 3 days to allow for best targeting and expression. Scale bar: 500 μm. (**d**) Sketch of behavior paradigm. Prior to behavioral testing and calcium imaging, mice were handled and habituated to the microscope and baseline (BL) environment. Each recording day began with a BL recording followed by a test with at least 2 hr between each session. On days with multiple test sessions (recall), the order of context (context A or context C) was counterbalanced. BL = Baseline, ctx A = context A, ctx C = context C. (**e**) Example raw traces of changes is calcium extracted with CNMF-E. Each line represents one neuron in the field of view. The lower trace shows examples of calcium events (red line) extracted from a raw trace.

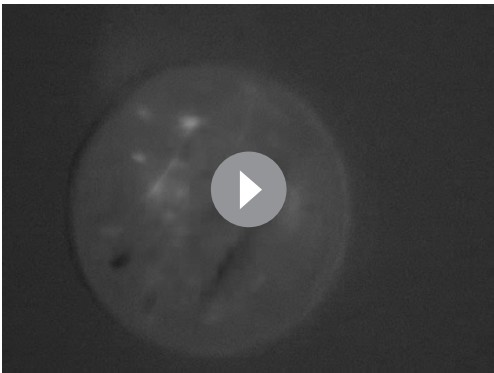

**Video 1.** Example recording of calcium activity in ACC neurons in lenti-shNT injected mice. Normalized (dF/F) calcium signals acquired during a 6-min baseline recording. This example video was spatially (4 x) and temporally (2 x) downsampled.
https://elifesciences.org/articles/70586/figures#video1

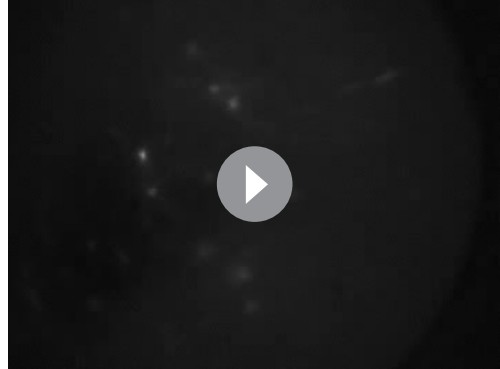

**Video 3.** example recording of calcium activity in CA1 neurons in lenti-shNT injected mice. Normalized (dF/F) calcium signals acquired during a 6-min baseline recording. This example video was spatially (4 x) and temporally (2 x) downsampled.
https://elifesciences.org/articles/70586/figures#video3

[One-sample t-test against 0; ACC shNT: N.S, n = 5, ACC shRNA: N.S., n = 5, CA1 shNT: N.S., n = 5 mice, CA1 shRNA: N.S. n = 5]. Finally, we performed the same analysis on the same day baseline recording as a control recording that did not include a foot-shock. We found a similar number of neurons that altered their activity similar to FSR neurons (shamFSR) in control mice (*Figure 2—figure supplement 1c*) while the shRNA group showed significantly less shamFSR neurons in ACC compared to FSR neurons found during the training session [Two-tailed paired t-test, ACC shNT: N.S., shRNA: p = 0.021]. There was no overall increase in event rate or correlated pairs in shamFSR neurons in either group or brain region (*Figure 2—figure supplement 1d-e*) [Paired t-test, event rate (d), ACC, shNT, N.S, p = 5, shRNA, N.S., n = 5; CA1, shNT, N.S., n = 5; shRNA, N.S., n = 5; correlated pairs (e), CA1, shNT, N.S., n = 5, shRNA, N.S., n = 5; CA1, shNT, N.S., n = 5, shRNA, N.S., n = 5].

Together these data suggest that increased DG – CA3 FFI facilitates foot-shock-induced neuronal responses and FSR ensemble-like activity in CA1 and ACC through increased recruitment of FSRs and synchronous activity of FSRs (*Figure 2i*).

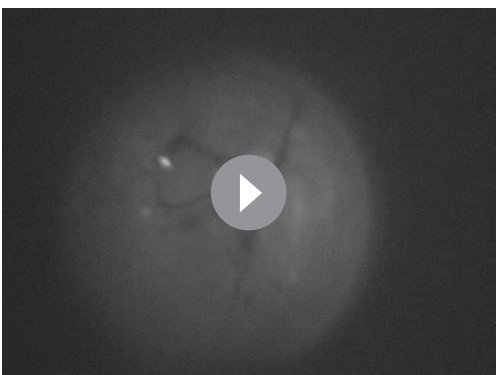

**Video 2.** Example recording of calcium activity in ACC neurons in lenti-shRNA injected mice. Normalized (dF/F) calcium signals acquired during a 6-min baseline recording. This example video was spatially (4 x) and temporally (2 x) downsampled.
https://elifesciences.org/articles/70586/figures#video2

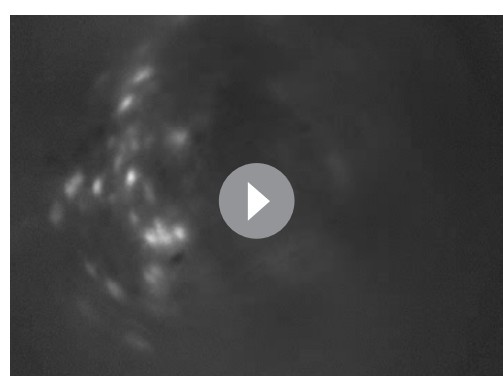

**Video 4.** example recording of calcium activity in CA1 neurons in lenti-shRNA injected mice. Normalized (dF/F) calcium signals acquired during a 6-min baseline recording. This example video was spatially (4 x) and temporally (2 x) downsampled.
https://elifesciences.org/articles/70586/figures#video4

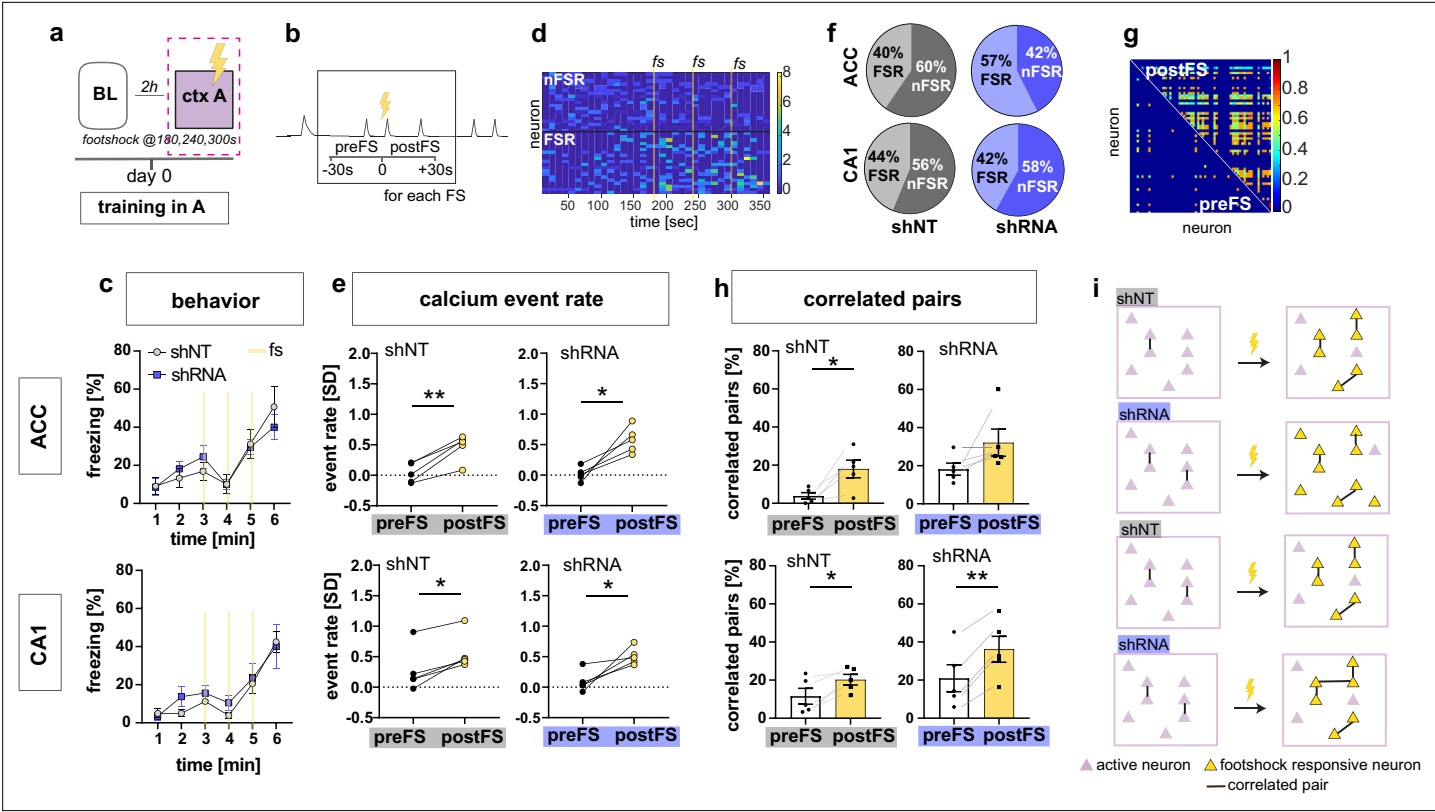

**Figure 2.** Increasing FFI promotes neuronal responses in CA1 and ACC during encoding of foot-shock. (**a**) Schematic of time point in behavioral paradigm of the presented data (context A on training day with three foot-shocks (FS) applied at 180, 240, and 360 s). (**b**) schematic of analysis window for neuronal activity before FS (preFS) and after FS (postFS) presented in panels e, g and h. (**c**) Quantification of FS response behavior measured as freezing in percentage of one minute time bins. All experimental groups showed a similar increase in freezing upon the FS training in context A with no main treatment effect [upper panel, ACC: Two-way repeated measures ANOVA, time x treatment effect p = 0.702: time main effect p < 0.0001, treatment main effect = 0.796, n = 5; lower panel CA1: time x treatment effect p = 0.838, time main effect p < 0.0001, treatment main effect = 0.49, n = 5 shNT, n = 6 shRNA]. (**d**) Example heatmap of z-scored neuronal activity (calcium events) in CA1 (shNT) across time in context A grouped by neuronal activity (foot-shock responsive, FSR; non-foot-shock responsive, nFSR). Colorbar represents SD (z-scored to first 180 s of the recording). (**e**) Neuronal activity of FSR neurons increased significantly after the FS was applied in all experimental groups. Neuronal activity was z-scored (over the first 180 s in context A) and averaged over the three foot-shocks using a 30 s window as preFS or postFS condition (see sketch in panel b) [Two-tailed paired t-test, ACC shNT preFS vs postFS: p = 0.004 (upper left); ACC shRNA preFS vs postFS: p = 0.013 (upper right), CA1 shNT preFS vs postFS: p = 0.007 (lower left), CA1 shRNA preFS vs postFS: p = 0.024 (lower right), n = 5]. (**f**) Number of FSR and nFSR neurons in each experimental group. In ACC, lenti-shRNA injection increased the number of FSR neurons (ACC, shNT, FSR, mean = 40.24, SEM = 4.06, nFSR, mean = 59.77, SEM = 4.06; ACC shRNA, FSR, mean = 57.43, SEM = 5.73, nFSR, mean = 42.57, SEM = 5.73; CA1, shNT, FSR, mean = 43.79, SEM = 4.35; nFSR, mean = 56.21, SEM = 4.35; shRNA, FSR, mean = 41.90, SEM = 3.47, nFSR, mean = 58.10, SEM = 3.47) (see also *Figure 2—figure supplement 1* for bar graphs of FSR and statistical analysis. (**g**) Example heatmap of pairwise correlation between neuronal pairs in CA1 before the FS (preFS, lower triangle) and after the FS (postFS, upper triangle) from one shNT injected mouse.). Colorbar represents correlation coefficient. (**h**) Number of correlated pairs (> mean + 2 SD from a shuffled control) among FSR neurons as percentage of all possible pairs of FSR neurons. All experimental groups, except ACC shRNA, showed an increase in the percentage of correlated pairs after the foot-shock. [Two-tailed paired t-test, ACC shNT: p = 0.021 (upper left), ACC shRNA: p = 0.141 (upper right), CA1 shNT: p = 0.021 (lower left), CA1 shRNA: p = 0.001 (lower right), n = 5]. (**i**) Graphic summary of findings presented in this panel. In both ACC and CA1, learning, in form of experiencing a foot-shock in context A, induced the formation of a neural ensemble among FSR neurons by increasing co-activity. Increased FFI facilitated this effect in both brain regions with a higher number of FSR neurons in ACC and a stronger increase in correlated pairs in CA1 (see also *Figure 2—figure supplement 1*). Throughout, statistical data are represented as mean ± SEM, *p < 0.05, **p < 0.01, ***p < 0.001, n represents number of mice.

The online version of this article includes the following figure supplement(s) for figure 2:

**Figure supplement 1.** Supporting data for *Figure 2*.

## FFI in DG − CA3 facilitates emergence of training context-specific ensembles in CA1 and ACC following learning

We next asked how increasing FFI modifies CA1 and ACC ensemble properties during recent recall of contextual fear memory. One day following fear conditioning, mice were returned to the conditioned context A and a novel, neutral distinct context C (*Figure 1a, d* and *Figure 3a*). Behavioral analysis revealed that both shRNA and shNT groups exhibited high levels of discrimination of the two contexts and showed similar levels of freezing in training context A and negligible levels of freezing in context C [Two-way repeated measures ANOVA, treatment x time effect N.S., time main effect p = 0.014; treatment main effect N.S.; n = 10, pooled ACC and CA1 mice. see *Figure 3—figure supplement 3c* for separate ACC and CA1 data] (*Figure 3b*). In context A, we observed synchronous activity, a learning-induced network property (*Gonzalez et al., 2019*; *Liu et al., 2017*; *Modi et al., 2014*; *Rajasethupathy et al., 2015*), suggestive of a context-associated ensemble in CA1. Specifically, both virus groups showed increased numbers of correlated neurons compared to baseline levels only in context A and not in the novel, distinct neutral context C (yet non-significant in shRNA mice) [one sample t-test against 1 (BL), context A, shNT: p = 0.003, n = 5, shRNA: N.S. (strong trend, p = 0.055), n = 5; context C: shNT: N.S., n = 5, shRNA: N.S., n = 5] (*Figure 3c–d*). Additionally, in context A, the shRNA group also showed a significant increase in number of correlated pairs in ACC indicative of emergence of a training context-associated ensemble in ACC [one sample t-test against 1 (BL), context A, ACC, shNT: N.S., shRNA: p = 0.018, n = 5] (*Figure 3c–d*) but we did not detect a difference between virus groups [two tailed unpaired student's t-test with welch correction for ACC context A: p = N .S, n = 5]. Using CellReg (*Sheintuch et al., 2017*) to register neurons across sessions, we found no reactivation of FSR neurons among the correlated pairs of neurons identified in CA1 in the recall session [shNT: mean: 0.0 S.E.M. 0; shRNA: mean 0.0, S.E.M, 0] (data not shown). We controlled for the possible effect of locomotion on neuronal activity by quantifying the time mice moved in each context and found no correlation between the number of correlated neuronal pairs and the time mice moved in the respective context [Pearson's correlation, ACC context A, $R^2$ = 0.19, p = 0.196, ACC context C: $R^2$ = 0.00, p = 0.935, CA1 context A: $R^2$ = 0.09, p = 0.376, CA1 context C: $R^2$ = 0.04, p = 0.599, n = 5 mice per group] (*Figure 3—figure supplement 1a-d*).

Analysis of constituent cell properties within recorded cell populations revealed that the majority of neurons were active in only one context with a small subset (CA1, shNT 15.4 ± 1.9, shRNA 14.1 ± 2.2; ACC, shNT 23.7 ± 2.4, shRNA 14.5 ± 4.6; percentage of A&C neurons expressed as mean ± S.E.M) of neurons active in both contexts (hereafter referred to as 'A&C neurons') (*Figure 3e–h*). We used a normalized jaccard similarity index (*Ahmed et al., 2020*), which corrects for differences in number of overall recorded neurons per session to calculate the average number of A&C neurons per day. Neurons in CA1 (including A&C neurons) showed higher activity i.e. rate of calcium transients in context C relative to context A [Two-way ANOVA with repeated measures, CA1, shNT: context x time effect p < 0.0001, time main effect p = 0.007, context main effect p = 0.028, n = 5; shRNA,: context x time effect p < 0.0001, time main effect p = 0.012, context main effect p = 0.029, n = 5] [A&C neurons: one sample t-test against 0, CA1, shNT: p = 0.007, shRNA: p = 0.048, n = 5] suggestive of a novelty response (*Figure 3—figure supplement 1i-j*, *Figure 3f*). In contrast to the training session, higher neuronal activity in context C was not accompanied by increased number of correlated pairs. Interestingly, the shRNA group showed significantly fewer A&C neurons in CA1 indicating that increasing FFI in DG − CA3 reduced the number of neurons that were active in both contexts [Two-tailed unpaired t-test with Welch's correction, p = 0.048, n = 5] (*Figure 3h*).

Thus, early stages of memory consolidation are characterized by increased co-activity of neurons within CA1 as also observed previously (*Ognjanovski et al., 2017*) and FSR neurons found during encoding do not appear to participate in CA1 context-specific ensembles at recent recall. Furthermore, increasing FFI in DG − CA3 facilitated the emergence of training context-specific ensembles in CA1 and ACC suggestive of increased SWR − spindle coupling between CA1 − ACC during memory consolidation (*Maingret et al., 2016*; *Ognjanovski et al., 2017*; *Steadman et al., 2020*; *Xia et al., 2017*).

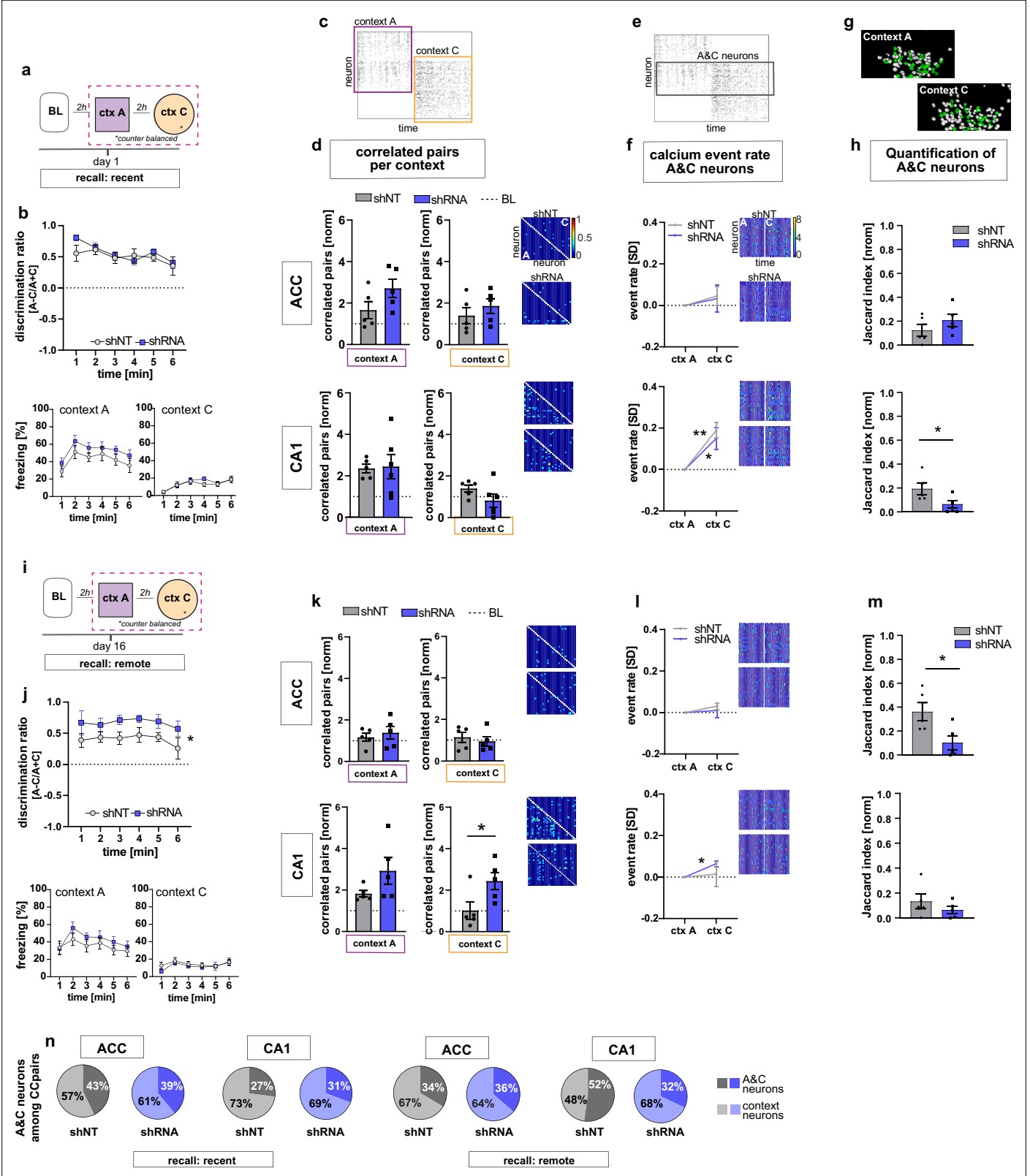

**Figure 3.** FFI in DG-CA3 dictates strength and specificity of CA1-ACC ensembles during memory consolidation. (**a**) Schematic of time point in behavioral paradigm of the presented data. (**b**) Quantification of discrimination ratio as relative difference in freezing in context A and context C (upper panel) and freezing time in context A (lower left) and context C (lower right). ACC and CA1 mice were grouped together by virus. One day after training, all mice discriminated between context A and C with no difference between treatment groups [Two-way repeated measures ANOVA, treatment x time

*Figure 3 continued on next page*

*Figure 3 continued*

effect p = 0.434, time main effect p = 0.014; treatment main effect p = 0.232; n = 10]. (**c**) Raster plot of calcium events over time in context A and context C after registering neurons across sessions. Each row corresponds to a neuron, x-axis corresponds to time in each context. Colored squares indicate the context the recording was acquired in. Empty rows in one context mean that the neuron was not active in this context. (**d**) Quantification of number of correlated pairs across all neurons per context. Context-specific neuronal pairs were analyzed by normalizing the correlated pairs to the baseline level recorded in the same day baseline recording (dotted line in graph). Heatmaps right to the graph show correlation values of example neurons in context A (lower triangle) and context C (upper triangle). In each graph, the upper heatmap shows an example of a lenti-shNT injected, the lower heatmap of a lenti-shRNA injected mouse. In CA1, lenti-shNT and lenti-shRNA injected mice showed a higher number of correlated pairs compared to baseline in context A (yet non-significant in shRNA mice) but not in the new context C [one sample t-test against 1, context A, shNT: p = 0.003, shRNA: p = 0.055, context C: shNT: p = 0.089, shRNA: p = 0.574, n = 5] with no difference between groups [Unpaired t-test with Welch's correction, context A: p = 0.889, n = 5, context C: p = 0.147, n = 5]. In ACC, lenti-shRNA injected mice showed a significantly higher number of correlated pairs in context A compared to baseline [one sample t-test against 1, context A, shNT: p = 0.188, shRNA: p = 0.018; context C: shNT: p = 0.368, shRNA: p = 0.073, n = 5] yet no difference was found between groups [Unpaired t-test with Welch's correction, context A: p = 0.120, n = 5, context C: p = 0.403, n = 5]. (**e**) Raster plot of calcium events over time in context A and context C after registering neurons across sessions (same as in panel c). Each row corresponds to a neuron. The square marks neurons that were active in both contexts (A&C neurons). (**f**) Normalized activity of A&C neurons. Heatmaps show activity of example neurons in context A (left side) and context C (right side) in the first 180 s per context for better visualization. In each graph, the upper heatmap shows an example of a lenti-shNT injected, the lower heatmap of a lenti-shRNA injected mouse. A&C neurons showed a significantly higher activity in context C compared to context A in CA1 but not in ACC [one sample t-test against 0, ACC, shNT: p = 0.399, shRNA: p = 0.65; CA1, shNT: p = 0.007, shRNA: p = 0.048, n = 5]. (**g**) Example spatial maps of neurons (white circles) in context A and context C with green circles indicating neurons that were registered to be active in both sessions (A&C neurons). (**h**) Quantification of A&C neurons using normalized jaccard similarity index. In CA1, lenti-shRNA injected mice showed significantly less A&C neurons compared to lenti-shNT injected mice [Unpaired t-test with Welch's correction, p = 0.048, n = 5]. No difference was found in ACC [Unpaired t-test with Welch's correction, p = 0.284, n = 5]. (**i**) Schematic of time point in behavioral paradigm of the presented data. (**j**) Quantification of discrimination ratio (see b for details). Sixteen days after foot-shock training in context A, both treatment groups discriminated between A and C. Lenti-shRNA injected mice showed a significantly higher discrimination ratio compared to lenti-shNT injected mice [Two-way repeated measures ANOVA, treatment x time effect p = 0.993, time main effect p = 0.412, treatment main effect: p = 0.028, n = 10, pooled CA1 and ACC mice]. (**k**) Quantification of number of correlated pairs across all neurons at day 16. Same presentation as in panel d. In CA1, both virus groups maintained an increased number of correlated neurons in context A compared to baseline [One-sample t-test against 1, context A, shNT, p = 0.006; shRNA, p = 0.04; n = 5]. Lenti-shNT mice showed a significantly lower number of correlated pairs compared to day 1 [comparison not shown, paired t-test with Welch's correction, p = 0.041, n = 5]. In context C, shRNA mice showed a similarly increased number of correlated neurons compared to baseline as in context A [One-sample t-test against 1, context C, shNT, p = 0.973; shRNA, p = 0.026; n = 5] with a significant difference between groups [Unpaired t-test with Welch's correction, p = 0.0423, n = 5]. In ACC, no difference was found between contexts [One-sample t-test against 1, context A, shNT, p = 0.444; shRNA, p = 0.285; context C, shNT, p = 0.609, shRNA, p = 0.803, n = 5]. (**l**) Normalized activity of A&C neurons (see **f**). In CA1, A&C neurons showed a significantly higher activity in context C compared to context A in shRNA mice [one sample t-test against 0, shNT: p = 0.824, shRNA: p = 0.012, n = 5]. No difference was found in ACC [one sample t-test against 0, shNT: p = 0.148, shRNA: p = 0.784, n = 5]. (**m**) Quantification of A&C neurons (see **h**). In ACC, lenti-shRNA injected mice showed significantly less A&C neurons compared to lenti-shNT injected mice [Unpaired t-test with Welch's correction, p = 0.028, n = 5]. No difference was found in CA1 [Unpaired t-test with Welch's correction, p = 0.340, n = 5]. (**n**) Quantification of A&C neurons that formed correlated pairs with any other neurons in shNT (grey graphs) and shRNA (blue graphs) injected mice during recent (left panel) and remote (right panel) recall. In CA1, lenti-shRNA injected mice showed fewer A&C neurons among the correlated pairs compared to lenti-shNT injected mice during remote recall [A&C neurons, recent, ACC, shNT 42.97 ± 8.37, shRNA 39.07 ± 5.93, CA1, shNT, 27.03 ± 8.23, shRNA, 30.64 ± 4.65; remote, ACC, shNT 33.59 ± 11.82, shRNA, 36.34 ± 12.14, CA1, shNT, 52.26 ± 4.80, shRNA, 28.55 ± 6.17; mean ± SEM] (see *Figure 3—figure supplement 1* for statistical analysis).

The online version of this article includes the following figure supplement(s) for figure 3:

**Figure supplement 1.** Supporting data for *Figure 3*.

**Figure supplement 2.** Supporting data for *Figure 3* (cell registration).

**Figure supplement 3.** Analysis of behavioral data during training (a, time freezing) or recall (**b–d**) to test for difference between control groups (ACC shNT vs CA1 shNT).

## FFI in DG − CA3 promotes stability and context specificity of CA1 − ACC ensembles at remote timepoint

Neuronal ensembles are thought to re-organize over time in the ACC as memories transition from a recent to a remote state. To determine how FFI in DG − CA3 influences properties of neuronal ensembles over time, we returned mice to the conditioned context A or neutral context C at 16 days following training and recorded neural activity in CA1 and ACC and measured freezing behavior (*Figure 3i and j*). Behavioral analysis revealed that the shRNA group exhibited significantly higher levels of discrimination between the two contexts than the shNT group [Two-way repeated measures ANOVA, treatment x time effect N.S., treatment main effect: p = 0.028, n = 10, see *Figure 3—figure supplement 3d* for comparison of ACC and CA1] (*Figure 3j*). Both groups of mice maintained an

increase in correlated pairs in CA1 during recall in context A (two-tailed unpaired student's t-test with Welch's correction, N .S., n = 5), however the shRNA group did not exhibit a significant decrease in correlated pairs from day 1 to day 16 as seen for shNT group (Two-tailed paired t-test, shNT p = 0.041, shRNA: N.S. n = 5, comparison not shown). Furthermore, the shRNA group showed increased co-activity in context C [One-sample t-test against 1 (baseline, BL), context A, shNT, p = 0.006; shRNA, p = 0.04; context C, shNT p = N .S.; shRNA, p = 0.026; n = 5;] that was significantly higher compared to shNT group in context C [two-tailed unpaired student's t-test with welch correction, p = 0.042, n = 5] (*Figure 3k*).

Next, we analyzed A&C neurons in CA1 and ACC. We found that the A&C neurons in CA1 in the shRNA group exhibited higher activity (rate of calcium events) in context C relative to context A [one sample t-test against 0, shNT: N.S., shRNA: p = 0.012, n = 5] (*Figure 3l*). This elevation in activity was peculiar to A&C neurons in shRNA group and was not detected in the overall activity of CA1 neurons in each context as seen on day 1 [Two-way ANOVA with repeated measures, CA1, shNT, context x time effect p,0.0001, context main effect N.S., time main effect: p = 0.0004, shRNA, context x time effect p < 0.0001, context main effect N.S., time main effect p = 0.005, n = 5] (*Figure 3—figure supplement 1k-l*). Furthermore, the shRNA group showed significantly fewer A&C neurons among all correlated pairs of neurons in CA1 compared to the shNT group [two-tailed unpaired t-test with Welch's correction, p = 0.017, n = 5; A&C neurons, recent, ACC, shNT 42.97 ± 8.37, shRNA 39.07 ± 5.93,CA1, shNT, 27.03 ± 8.23, shRNA, 30.64 ± 4.65; remote, ACC, shNT 33.59 ± 11.82, shRNA, 36.34 ± 12.14, CA1, shNT, 52.26 ± 4.80, shRNA, 28.55 ± 6.17; mean ± SEM] (*Figure 3n* and *Figure 3—figure supplement 1g-h*). While there was no difference in the number of A&C neurons between virus groups in CA1 [two tailed unpaired t-test with Welch's correction, N.S., n = 5], the shRNA group showed significantly fewer A&C neurons in ACC indicating that fewer neurons were active in both contexts C [two-tailed unpaired t-test with Welch's correction, p = 0.028, n = 5] (*Figure 3m*). The control group showed an increase in the number of A&C neurons over time (Two-way repeated measures ANOVA, treatment x time effect p = 0.017, Sidak's multiple comparison test day 1 vs day 16, shNT p = 0.039, shRNA p = 0.405, comparison not shown).

Given that identification of A&C neurons may vary based on the approach used to register cells (*Sheintuch et al., 2017*), we tested if the certainty of registration or the method of registration accounted for the virus-mediated effects that we found. We found no significant correlation between the registration certainty of the mice included in this study and the number of A&C neurons that could explain our virus-mediated effects (Pearson's correlation, shNT, ACC, $R^2$ = 0.09, p = 0.41, n = 10, shNT, CA1, $R^2$ = 0.05, p = 0.520, n = 10, shRNA, ACC, $R^2$ = 0.036, p = 0.600, n = 10, shRNA, CA1,

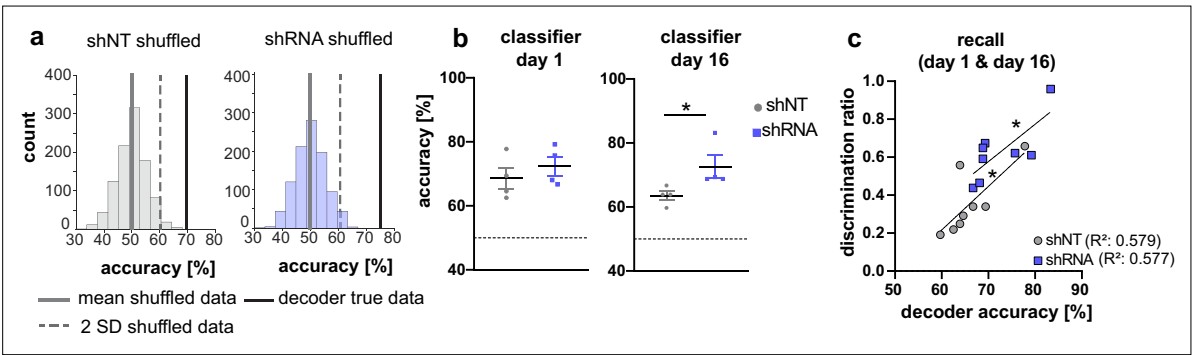

**Figure 4.** Activity of A&C neurons in CA1 conveys contextual information. (**a**) A support vector machine was trained to classify the context based on neuronal activity of CA1 A&C neurons. A distribution of prediction accuracy was created with 1,000 randomized datasets of shuffled event times (shuffled). Graphs show an example distribution of an shNT-injected (left) and an shRNA-injected mouse (right) recent recall (day 1). Lines in graph indicate the mean (solid, gray) and two standard deviations from the mean (dashed, gray). The solid line (black) indicates the decoder accuracy of the true data from the respective mouse. (**b**) Summary of prediction accuracy (within subject) for day 1 and day 16. On day 1, the accuracy to predict the context based on A&C neuron's activity was above chance (also see shuffled data in panel a) [One-sample t-test against 50 (chance level)], shNT, p = 0.012, shRNA, p = 0.005, n = 4 (only mice with >50 A&C neurons were included to avoid overfitting) with no difference between the treatment groups [Mann Whitney test, p = 0.486, n = 4]. This was maintained on day 16 [One-sample t-test against 50 (chance level), shNT, p = 0.003, shRNA, p = 0.008, n = 4] with a significantly higher accuracy in lenti-shRNA injected mice [Mann Whitney test, p = 0.029, n = 4]. (**c**) Accuracy of the decoder correlates positively with the discrimination ratio found in mice during recall (day 1 and day 16) in both groups. Pearson's $R^2$ value is shown in brackets. [Pearson's correlation, shNT p = 0.028, shRNA = 0.028, n = 4 per group].

$R^2$ = 0.029, p = 0.638, n = 10) (*Figure 3—figure supplement 2a-b*). When we used a fixed threshold, instead of a probability model, to register cells across sessions, we were able to reproduce the virus-mediated differences seen in CA1 at a recent time point as well as in ACC at the remote timepoint demonstrating the robustness of our findings (*Figure 3—figure supplement 2c*).

Together, these data suggest that increasing FFI in DG – CA3 prevents time-dependent decay of training context-associated ensemble in CA1, promotes formation of a neutral context-associated ensemble in CA1 and emergence of context-specific ensembles in ACC.

## A&C neuron activity in CA1 is sufficient to accurately predict context

Our analysis of CA1 and ACC ensembles at recent and remote timepoints suggests a sequential establishment of context-specific ensembles in CA1 followed by formation of context-specific ensembles in ACC at remote timepoint. While the majority of neurons were found to be only active in one context, our data suggest that the activity of A&C neurons in CA1 may also convey context-specific information. (*Figure 3f and l*). Therefore, we asked whether the activity of A&C neurons in CA1 is sufficient for a decoder to predict context. A support vector machine trained on time-binned events (5 s) in A&C neurons in CA1 performed better than shuffled data (see example distribution in *Figure 4a*). At the recent time point, day 1, calcium events in A&C neurons in both groups were sufficient to reliably predict context [Mann Whitney test, N .S., n = 4; One-sample t-test against 50 (chance level), shNT, p = 0.012, shRNA, p = 0.005, n = 4]. The shRNA group maintained a similarly high level of information in the A&C neurons as we were able to predict the context with similar accuracy at the remote time point [One-sample t-test against 50 (chance level), shRNA, p = 0.008, n = 4]. In contrast, decoder trained on shNT mice showed a reduction in prediction accuracy for the remote time point resulting in significantly less prediction accuracy in shNT mice compared to shRNA mice [Mann Whitney test, p = 0.029, n = 4] (*Figure 4b*). Nonetheless, the prediction accuracy in shNT mice maintained a significant-above-chance performance despite the fact that A&C neurons did not show an elevation in activity at the remote time point (*Figure 3l*) [One-sample t-test against 50, shNT, p = 0.003, n = 4] (*Figure 4b*) supporting that both activity levels and synchronous firing are likely determinants of context specificity. We further tested if the decoder accuracy correlated with the ability of mice to discriminate between the two contexts. We found a positive correlation between these two parameters indicating that A&C neurons may indeed serve to guide context-associated behavioral responses [Pearson's correlation, shNT: $R^2$ = 0.579, p = 0.028, n = 8, shRNA $R^2$ = 0.577, p = 0.028, n = 8].

## Enhancing FFI in DG – CA3 promotes cross-region communication in CA1 – ACC network

Our findings that enhancing FFI in DG – CA3 FFI facilitated the formation of context associated neuronal ensembles in both CA1 and ACC strongly suggest a role for increased hippocampal-cortical communication. To directly test this hypothesis, we performed simultaneous recordings of local field potentials (LFP) in CA1 and ACC using custom made tetrodes in mice following viral shNT/shRNA injections into the DG (*Figure 5a*). We analyzed ripples in CA1 and spindles in ACC during slow-wave sleep (SWS) as extensive work has implicated ripple – spindle coupling in memory consolidation (*Battaglia et al., 2004*; *Maingret et al., 2016*; *Sirota et al., 2003*; *Xia et al., 2017*). Recordings were performed in a familiar context before and after foot-shock training in context A (*Figure 5b*). All mice showed similar duration of SWS (*Figure 5—figure supplement 1a*) [Two-way repeated measures ANOVA, treatment x time effect, N.S., main treatment effect, N.S., main time effect, N.S, n = 5 per group].

In learning naïve shRNA-injected mice, we found significantly more ripples that co-occur with spindles (coupled ripples) compared to control mice (*Figure 5c*, Two-tailed unpaired student's t-test with Welch's correction, p = 0.049, n = 5 per group). This was not due to an increase in ripple or spindle occurrence (*Figure 5d–e*) [Two-tailed unpaired student's t-test with Welch's correction, ripples/sec: N.S., n = 5 per group, spindles/sec, N.S., n = 5]. The number of overall detected ripple or spindle events did also not differ between groups (*Figure 5—figure supplement 1b,c*) [total number of spindles: Two-way repeated measures ANOVA, treatment x time effect, N.S., main treatment effect, N.S., main time effect, N.S., n = 5 per group; total number of ripples: Two-way repeated measures ANOVA, treatment x time effect, N.S., main treatment effect, N.S., main time effect, N.S., n = 5 per group].

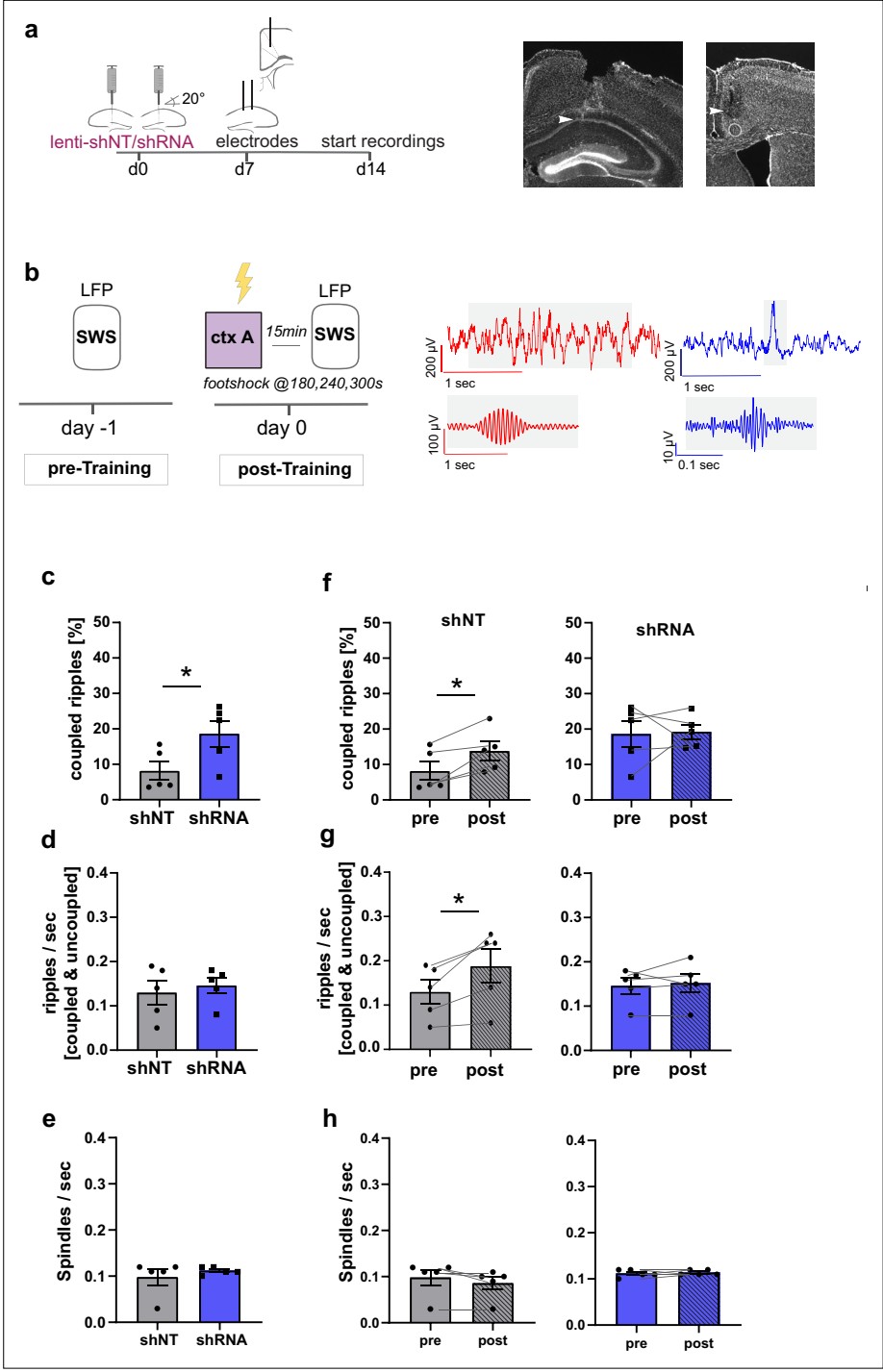

**Figure 5.** Increased FFI in DG – CA3 enhances CA1 ripple – ACC spindle coupling. (**a**) Schematic of surgical procedure and histology of tetrodes in CA1 and ACC (white arrows). (**b**) Recording paradigm and example traces. Local field potentials (LFPs) were recorded before and after mice were trained in context A. Raw traces obtained from LFP recordings from ACC (red) and CA1 (blue). Gray boxes indicate time points magnified in filtered examples below. Magnified signals were filtered for ripples (CA1, blue) and spindles (ACC, red) during slow-wave sleep (SWS). Magnified examples are not aligned temporally. (**c**) Coupled ripples (% of all ripples) during pre-Training recordings. shRNA mice showed significantly more coupled ripples compared to shNT mice [Unpaired Student's t-test, p = 0.049, n = 5 per group]. (**d-e**) Ripple occurrence (ripples/sec) and spindle occurrence (Spindles/sec) during pre-Training recordings. No difference was found between virus groups. [Unpaired Student's t-test, ripples/sec: p = 0.633, n = 5 per group; spindles/sec: p = 0.466, n = 5 per group]. (**f–h**) Effect of training on coupled

*Figure 5 continued on next page*

*Figure 5 continued*

ripples, ripple occurrence and spindle occurrence. Plots show pairwise comparison of pre-Training (same data as c–e) with post-Training recordings. shNT mice (left panel) show increased coupled ripples (**f**) following learning [Paired Student's t-test, p = 0.019, n = 5; shRNA mice: p = 0.881, n = 5] and increased couple occurrence (**g**) [Paired Student's t-test, p = 0.031, n = 5; shRNA mice: p = 0.621, n = 5]. No difference is found in spindle occurrence following learning (**h**) in either group [Paired Student's t-test, p = 0.109, n = 5; shRNA mice: p = 0.623, n = 5].

The online version of this article includes the following figure supplement(s) for figure 5:

**Figure supplement 1.** Supporting data for *Figure 5*.

We next quantified the effect of contextual fear learning on ripple – spindle coupling. We recorded LFPs 15 minutes after mice were returned to the familiar context from training context. In line with previous studies, we found an increase in coupled ripples in control mice (*Maingret et al., 2016*; *Xia et al., 2017*) [Paired Student's t-test, p = 0.019, n = 5] to a similar level as seen in naive shRNA-injected mice (*Figure 5f*). Thus, our viral manipulation promotes the reorganization of neuronal activity patterns in downstream CA1 ACC that naturally occurs following learning. While training induced an increase in overall ripple occurrence (ripple/sec, *Figure 5g*) in control mice [Paired Student's t-test, p = 0.031, n = 5], this was not seen in shRNA mice [Paired Student's t-test, N.S., n = 5].

## Formation, stabilization and specificity of context-associated ensembles

Our longitudinal imaging studies reveal evolving dynamics of ensemble properties in CA1 – ACC networks underlying recent and remote memories and demonstrate how enhancing FFI in DG – CA3 impacts these properties to promote memory consolidation. Learning induced the formation of neuronal ensembles of foot-shock-responsive neurons in both ACC and CA1 and enhanced FFI potentiated this property in CA1. During early stages of memory consolidation (recall of recent memory), we observed the emergence of training context-associated ensembles (increased numbers of co-active neurons) in both CA1 and ACC. Increasing FFI in DG – CA3 prior to learning promoted emergence of training context-associated ensemble in ACC suggestive of increased CA1 – ACC communication (*Figure 6a*). Additionally, we observed a FFI dependent reduction in the number of neurons active in both training and neutral contexts in CA1 (*Figure 6b*, top) reflecting increased specificity of neuronal ensembles. At remote recall, we observed that FFI in DG – CA3 prevented time-dependent decay of the training context-associated ensemble and promoted acquisition (or maintenance since it may have emerged prior to test at remote timepoint) of a neutral context specific ensemble in CA1 (*Figure 6a and b*). Within the ACC, and not evident at the recent timepoint, we observed a FFI dependent reduction in the number of neurons active in both training and neutral contexts (*Figure 6b*, bottom). Electrophysiological recordings suggest that increasing FFI in DG – CA3 induced learning-dependent enhancement in network synchrony in CA1 – ACC in behaviorally naïve mice (*Figure 6c*).

## Discussion

The DG – CA3 circuit contributes to encoding of episodic memories conjunctive representations that relate people, objects, events with the spatial and temporal contexts in which they occur. This is thought to be accomplished in DG – CA3 by integration of processed sensory information about our external world channeled from higher order cortices (perirhinal, postrhinal) via the entorhinal cortex and by decreasing interference between similar representations (*Hainmueller and Bartos, 2020*). Distinct or updated detail-rich contextual representations in CA3 are transferred to CA1 enroute to prefrontal cortical sites (ACC, infralimbic and prelimbic) for memory consolidation. PV INs in CA1 and ACC are thought to mediate learning-induced synchronization of neuronal activity to stabilize memory ensembles and channel information across different regions (*Buzsáki, 2015*; *Çaliskan et al., 2016*; *Dupret et al., 2013*; *Fernández-Ruiz et al., 2021*; *Gan et al., 2016*; *Ognjanovski et al., 2017*; *Stark et al., 2014*; *Xia et al., 2017*) We know much less about how PV INs in DG – CA3 contribute to these ensemble dynamics during memory consolidation.

To address this gap in our knowledge, we deployed a previously characterized molecular tool to increase dentate granule cell recruitment of PV IN mediated perisomatic inhibition onto CA3 in a way that mimics learning induced modifications of dentate granule cell – PV IN – CA3 connectivity

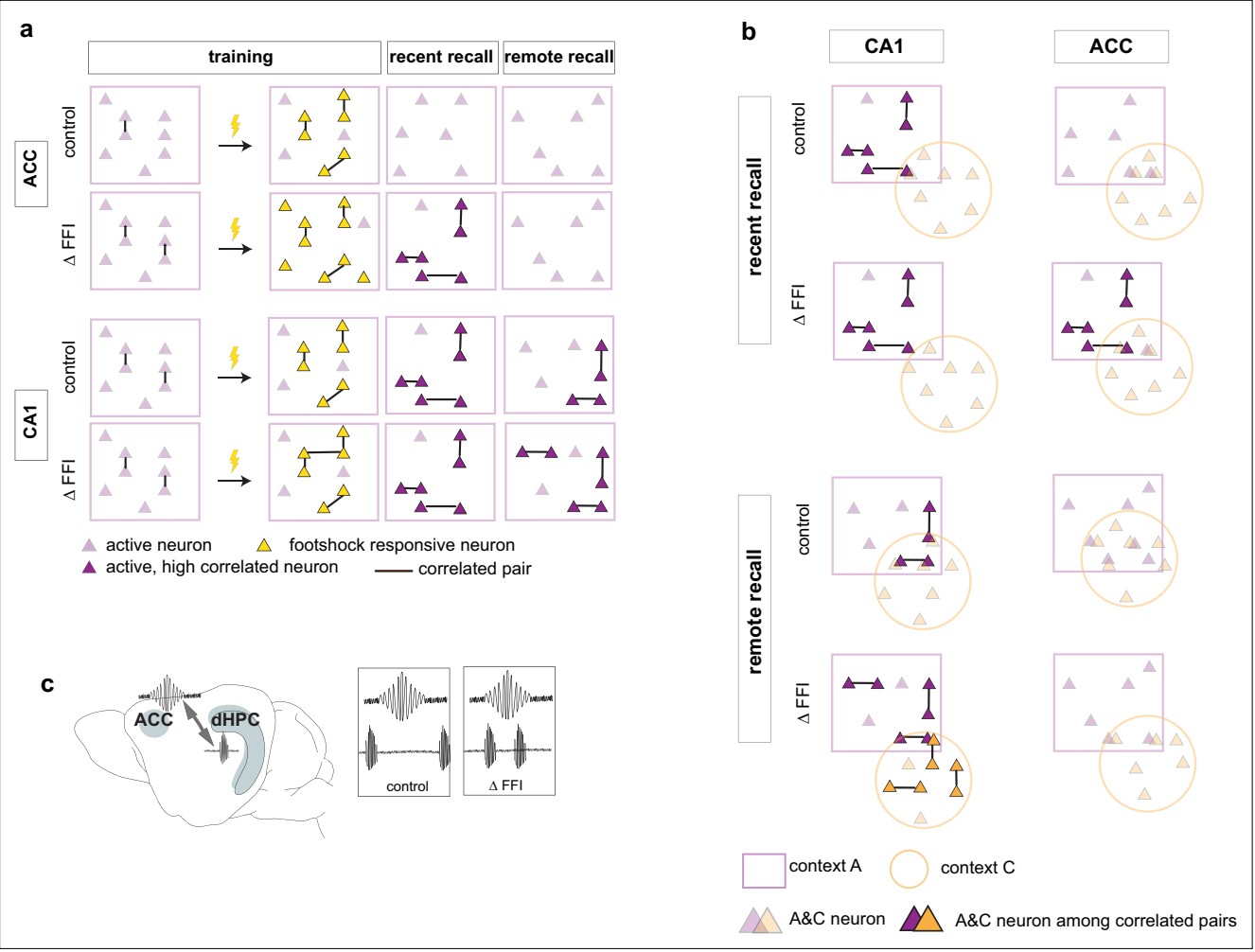

**Figure 6.** Increased FFI in DG-CA3 promotes systems consolidation in hippocampal-cortical networks. (**a–b**) Graphic summary of current findings. Learning induced the formation of neuronal ensembles of foot-shock-responsive neurons in both ACC and CA1 and FFI potentiated this property in CA1. During early stages of memory consolidation (recall of recent memory), we observed the emergence of training context-associated ensembles (increased numbers of co-active neurons) in both CA1 and ACC. Increasing FFI in DG-CA3 promoted emergence of training context-associated ensemble in ACC and increased CA1-ACC communication (**c**). Additionally, we observed a FFI dependent reduction in the number of neurons active in both training and neutral contexts in CA1 (**a**, top) reflecting increased specificity of neuronal ensembles. At remote recall, we observed that FFI in DG-CA3 prevented time-dependent decay of the training context-associated ensemble and promoted acquisition (or maintenance since it may have emerged prior to test at remote timepoint) of a neutral context specific ensemble in CA1 (**a**, **b**). Within the ACC, and not evident at the recent timepoint, we observed a FFI dependent reduction in the number of neurons active in both training and neutral contexts (**b**, bottom).

(**Guo et al., 2018**). By combining this molecular approach to enhance FFI in DG – CA3 with longitudinal calcium imaging of neuronal ensembles, we identified PV INs in DG – CA3 as critical arbiters of ensemble stabilization and specificity in CA1 – ACC networks during memory consolidation. Furthermore, we provide direct evidence for a role of PV INs in the DG – CA3 circuit in promoting hippocampal-cortical communication. Our approach affords a unique opportunity to understand how a defined inhibitory connectivity motif (dentate granule cell – PV IN – CA3), rather than PV INs in CA3 per se, contributes to evolving ensemble properties underlying memory consolidation. While optogenetic or chemogenetic manipulations of PV INs (in CA1) have illuminated their roles in regulation of synchronous activity and in SWR – spindle coupling, these manipulations target PV INs rather than a circuit motif in a wiring diagram and as such, do not address input-specific recruitment of PV INs in these processes. Here, we interpret our data on stability and specificity of ensembles in CA1 and ACC during memory consolidation through the lens of this inhibitory circuit motif in DG – CA3.

We found synchronous activity of principal neurons (captured in correlated pairs) indicative of neuronal ensembles in CA1 and ACC during training, recent and remote recall (**Gonzalez et al., 2019**;

*Liu et al., 2017*; *Modi et al., 2014*; *Rajasethupathy et al., 2015*). In contrast to a prior report on FSR neurons in ventral CA1 that project to the basolateral amygdala (*Jimenez et al., 2020*), we did not observe recruitment of FSR neurons into training context-associated ensembles in CA1 during recent recall. This differential contribution of dorsal and ventral hippocampal FSR neurons to context-associated ensembles may reflect output connectivity (basolateral amygdala vs. other downstream circuits) and distinct roles of dorsal and ventral hippocampus in encoding spatial and non-spatial information (*Chockanathan and Padmanabhan, 2021*; *Fanselow and Dong, 2010*; *Strange et al., 2014*).

Our data demonstrate that increased FFI in DG – CA3 promotes neuronal ensembles in both CA1 and ACC. Following contextual fear learning, we observed emergence of training context-associated ensembles in CA1 in both groups of mice but only mice with increased FFI in DG – CA3 exhibited a learning-induced ensemble associated with the training context in the ACC. Analysis of neuronal ensembles during remote recall demonstrated that increasing FFI in DG – CA3 prevented the decay of training context associated ensemble and promoted the emergence of a neutral context associated ensemble in CA1.

It is thought that the DG – CA3 – CA1 – ACC network undergoes changes in synaptic weights following learning that likely result in inter-regional ensembles. During memory consolidation, those neuronal representations are reorganized and strengthened potentially through re-activation during sharp-wave ripples and transferred to cortical regions by coupling between ripples and cortical spindles (*Cheng and Frank, 2008*; *Hwaun and Colgin, 2019*; *Maingret et al., 2016*; *Makino et al., 2019*; *Sasaki et al., 2018*; *Xia et al., 2017*). Our manipulation is restricted to the DG – CA3 circuit and thus strongly implicates FFI in DG – CA3 in formation and maintenance of CA1 and ACC ensembles. Previous work showed that increasing FFI in DG – CA3 decreases overlap between two context-associated ensembles in CA3 (*Guo et al., 2018*). We predict that increasing granule cell recruitment of perisomatic inhibition onto CA3 neurons imposes a sparser activity regimen from DG to CA3 thereby reducing the likelihood of recruitment of the same CA3 neurons to distinct ensembles during learning (*Csicsvari et al., 2000*; *de la Prida et al., 2006*; *Gómez-Ocádiz et al., 2021*; *Mori et al., 2007*; *Neubrandt et al., 2017*; *Sasaki et al., 2018*; *Torborg et al., 2010*). In turn, assemblies of CA3 neurons encoding different contexts entrain activity of downstream ensembles of CA1 principal neurons (*Choi et al., 2018*; *Csicsvari et al., 2000*; *de la Prida et al., 2006*). Distinct ensembles in CA1 recruit local PV INs to synchronize their activity and form SWRs (*Choi et al., 2018*; *Csicsvari and Dupret, 2014*; *Gan et al., 2016*; *Malerba et al., 2017*; *Ognjanovski et al., 2017*; *Stark et al., 2014*). CA3 neurons that encode novel spatial information were shown to be more likely active during SWR in CA1 suggesting that interregional maps may be strengthened during consolidation (*Hwaun and Colgin, 2019*). PV INs have been implicated to regulate fidelity of spike transfer (i.e. in form of coordinated spiking or burst firing of excitatory neurons) (*Lamsa et al., 2005*; *Mori et al., 2007*). Increased FFI inhibition may thus not only enhance sparseness but facilitate activation of CA3 ensembles resulting in improved memory consolidation in the DG – CA3 – CA1 circuit. Consequently, these changes in network properties of DG – CA3 – CA1 facilitate transfer of information to ACC via SWR – spindle coupling for memory consolidation (*Maingret et al., 2016*; *Xia et al., 2017*). Our findings of enhanced SWR – spindle coupling in mice with increased FFI in DG – CA3 provide direct evidence for such facilitated information transfer across the CA1 – ACC network.

Importantly, feedforward inhibition in DG – CA3 synapses is naturally enhanced for several hours following learning (see *Ruediger et al., 2011*; *Guo et al., 2018*). By leveraging a molecular tool to enhance FFI prior to learning, we were able to reconfigure DG – CA3 circuitry in a learning permissive state. This is perhaps why behaviorally naïve mice with increased FFI in DG – CA3 exhibit increased ripple-spindle coupling prior to learning.

Increasing FFI in DG – CA3 also enhanced specificity of context-associated ensembles in CA1 and ACC. Previous studies demonstrated that subsets of neurons are shared across ensembles when mice experience two distinct environments proximal in time (*Cai et al., 2016*; *Rubin et al., 2015*). We found a reduction in the number of neurons active in both contexts (A&C neurons) in ACC and a smaller proportion of A&C neurons within context-associated neuronal ensembles in CA1 at remote recall. These A&C neurons may function as "schema" neurons in that they represent shared features (statistical regularities) across ensembles and facilitate the formation of new context-associated ensembles (*Abdou et al., 2018*; *McKenzie et al., 2014*; *McKenzie et al., 2013*; *Tse et al., 2007*). Increased neuronal excitability has been shown to bias recruitment of neurons into ensembles (*Yiu et al., 2014*;

*Zhou et al., 2009*). Higher activity of A&C neurons (observed upon exposure to context C) may bias their allocation into the ensemble associated with the new context while maintaining context-specific activity through synaptic connectivity (*Abdou et al., 2018*; *Gava et al., 2021*). Indeed, we can train a decoder on activity of A&C neurons to predict context. The decoder accuracy positively correlated with the mice' ability to discriminate contexts which indicates that these neurons are behaviorally relevant. Computational modeling has suggested that CA1 balances abstraction of statistical regularities across memories mediated by the temporoammonic pathway (EC → CA1) with encoding of detail-rich distinct contextual representations conveyed through the trisynaptic EC → DG → CA3 → CA1 circuit (*Schapiro et al., 2017*). Our data supports this model since we observe a reduction in neurons encoding overlapping features (A&C neurons) in CA1 (this study) when we increase FFI in DG – CA3 and reduce interference between context associated ensembles in CA3 (*Allegra et al., 2020*; *Guo et al., 2018*; *Koolschijn et al., 2019*; *Leutgeb et al., 2004*). Thus, our manipulation biases the trade-off in CA1 towards generation of context-specific ensembles in CA1 over generation of schema. While shifting this balance in healthy animals may come at a cost, reversing reductions in FFI in DG – CA3 found during aging or Alzheimer's disease may represent a strategy to restore memory consolidation (*Guo et al., 2018*; *Viana da Silva et al., 2019*, Twarkowski and Sahay unpublished observations).

Our data builds on foundational theories (*McClelland et al., 1995*; *Nadel and Moscovitch, 1997*; *Teyler and DiScenna, 1986*; *Winocur et al., 2010*; *Winocur et al., 2007*) to suggest a continuous role for CA1 in playing an instructive teacher-like role in governing re-organization of ensembles in ACC during memory consolidation. During memory consolidation, conjunctive representations are thought to be transformed into more schema like or gist-like structures necessary for cognitive flexibility and new learning. However, the extent to which memories retain details may be determined by the hippocampus. Increasing FFI in DG – CA3 appears to facilitate re-organization of ensembles in ACC and decrease the number of A&C neurons (potentially, schema neurons) at the remote timepoint. Although in this study's experimental design, our behavioral analysis did not capture time-dependent generalization of the contextual fear memory, mice with increased FFI in DG – CA3 exhibited significantly greater long-term memory than controls. We infer from our data that stabilization of training-context associated ensembles in CA1 over time governs a toggle switch in the ACC that determines the extent to which memories may retain details over time. Such an interpretation is consistent with hippocampal indexing theory in that the maintenance of the hippocampal index for an experience permits access to details of the original experience stored in distributed neocortical sites (*Besnard and Sahay, 2016*; *Goode et al., 2020*; *Koolschijn et al., 2019*; *Teyler and DiScenna, 1986*).

A major challenge in neuroscience is to assign functional significance to circuit motifs in wiring diagrams. Here, we build on our past work to demonstrate how the identification of learning-induced molecular regulators of connectivity may illuminate the relationship between a DGC – PV IN – CA3 circuit motif with emergent network properties during memory consolidation. The discovery of other molecular specifiers of principal neuron-inhibitory neuron connectivity in combination with a 'bottom-up synapse to systems approach' as conveyed here is likely to generate new fundamental insights into the physiological relevance of distinct hippocampal inhibitory circuit motifs in memory processing. It is plausible that connectivity re-engineering strategies such as that described in this study harbor potential to enhance memory consolidation in aging and in mouse models of Alzheimer's disease (*McAvoy and Sahay, 2017*).

## Materials and methods
### Animal care
All mice were group housed in a 12 hr (7 am to 7 pm) light/dark cycle at room temperature with ad libitum access to food and water. After surgery, mice were split into pairs and housed with a divider to protect the implant without single housing mice. Male, 2–3 months old C57Bl6/J mice were obtained from Jackson Laboratories. All animals were handled, and experiments were conducted in accordance with procedures approved by the Institutional Animal Care and Use Committee at the Massachusetts General Hospital in accordance with NIH guidelines (IACUC 2011N000084).

Lenti virus production lenti-shNT and lenti-shRNA virus were generated in the lab using previously described pLLX-shRNA lentiviral vectors (*Guo et al., 2018*) or a similar lentivirus U6-based shRNA

knockdown vector from VectorBuilder. Expression of GFP or mCherry under the *ubiquitin C* promoter allowed to monitor the efficiency of transfection and infection both during virus production as well as in *in vivo* experiments. HEK293T cells were transfected with lentiviral vector and packaging plasmids, VSVG and Δ8.9, as described previously (*Guo et al., 2018*; *Lois et al., 2002*) to generate lenti-virus stock solutions. We used Lenti-X GoStix Plus (Takara Bio USA) to quantify the titer of each virus batch used in this study. The generated virus was further validated both *in vitro* by infection HEK293T cells and *in vivo* by injection into the hippocampus of 2–3 month old male C57BL6 mice. All virus stock solutions used in this study ranged from 200 to 300 ng/ml p24. Virus aliquots were stored at –80 °C for a maximum of 12 month and not reused (no freeze-thaw cycle).

## Virus injections and stereotactic surgeries

Prior to surgery mice received an injection of carprofen (5 mg/kg, subcutaneously, Patterson Veterinary Supply). Mice were anaesthetized with ketamine and xylazine (10 mg/mL and 1.6 mg/mL, intraperitoneally, Patterson Veterinary Supply) and placed at a planar angle in a stereotaxic frame (Stoelting). Eyes were protected with Puralube (Dechra). After exposing the skull, craniotomies above the target site for injections were created using a Foredom K.1070 High Speed Rotary Micromotor Kit. The injection was performed with a hamilton microsyringe (Hamilton, Neuros Syringe 7001) that was slowly lowered into the target location. For virus validation and calcium imaging of ACC neurons, bilateral lenti-virus injections into the DG were performed using following coordinates: –1.9 mm (AP),±1.35 mm (ML), –2.2 mm (DV) relative to bregma. For calcium imaging of CA1 neurons, we used an angle to access DG without damaging overlaying CA1. We adjusted the manipulator arm to an angle of –20° and used the following coordinates: –1.1 mm (AP),±1.4 mm (ML), –2.3 mm (DV) relative to bregma. A total of 0.5 µl virus solution (non diluted) was injected per hemisphere.

The calcium indicator GCaMP6f was unilaterally injected using an AAV1.CaMKII.GCaMP6f.WPRES. SV40 virus (Penn Vector Core) in either ACC ( + 1.0 mm (AP), –0.35 mm (ML), –2.0/–1.6 (DV)) or CA1 (–1.9 mm (AP), –1.4 mm (ML), –1.6 mm (DV)). For ACC, we spread the injection across two depths to achieve a better distribution of the virus. A total of 0.5 µl at a 1:5 dilution was injected. Mice that underwent calcium imaging in CA1, received first the GCaMP6f injection in CA1 and 3 days later the angled lenti-virus injection into the DG. Mice with ACC imaging received both injections on the same day. At least 10 min after injection, needles were removed, and the skin incision closed with coated vicryl sutures (Ethicon US LLC). Mice received a daily injection of carprofen (5 mg/kg, subcutaneously, Patterson Veterinary Supply) for 3 days following surgery.

Mice that underwent electrophysiological recordings received injection of lentivirus into the dDG as described above 1 week prior to implantation of electrodes. Custom made tetrodes (California Fine Wire Company, StablOhm 675, 0.003 inches) were implanted into ACC (AP: + 1.0, ML: –0.35, DV: –1.8) and CA1 (AP: –1.8, ML: –1.2 and –1.4 (2 tetrodes), DV: –1.6) for simultaneous recordings of local field potentials. Craniotomies were performed at the respective locations, the dura was perforated and the tetrodes slowly lowered into the tissue relative to bregma. A reference electrode (single wire) was placed in the cerebellum. A ground screw and an additional screw were implanted in the left hemisphere (Basi Bioanalytical Systems). Tetrodes and screws were fixed to the skull with superglue and dental cement (Stoelting).

## Lens implantation

At least one week after the last virus injection, mice were implanted with a GRIN lens (0.5 mm Ø x 4 mm in ACC, 1 mm Ø x 4 mm in CA1). We applied the same analgesia and anesthesia strategy as described above. Mice were placed in the stereotaxic frame and fixed at a planar angle. The skull was exposed, and a craniotomy created above the target site (ACC: + 0.9 mm (AP), –0.3 mm (ML), –1.3 mm (DV); CA1 –1.85 mm (AP), –1.35 mm (ML), –1.4 mm (DV)). In addition, we created two small craniotomies to insert anchor screws (Basi Bioanalytical Systems). For CA1, we removed the overlaying cortex, as described *Kinsky et al., 2020* using blunt needles of decreasing diameter (25ga to 31ga, SAI Infusion Technologies) attached to a vacuum pump while constantly applying cooled, sterile saline. A microscope (Nikon SMZ800) was placed above the stereotaxic frame to allow visual control of the procedure. Once the cortex was removed and possible bleeding stopped with gelfoam (Pfizer), the lens was attached to a custom-made lens holder and slowly lowered into the craniotomy. Coordinates for the lens were referenced to bregma. For ACC, no aspiration was performed. Instead,

we inserted a 30ga needle at the ACC coordinates (AP & ML) and slowly lowered it –1.0 mm DV into the cortex to open the track for the lens. After 10 min the needle was slowly removed and the GRIN lens, attached to a custom-made holder, inserted into ACC. Once the lens in either brain region was inserted, the gap between lens and skull was sealed with superglue before the lens was cemented to the skull with dental cement (Stoelting). A thin layer of blackened dental cement (by adding nail polish to the cement) was applied as final layer to shield the lens from light. The remaining part of the lens was covered with Kwik-cast sealant (World Precision Instruments).

One week following the lens implant, mice were placed back in the stereotaxic frame as described above. The Kwik-cast sealant was removed from the lens and possible residues cleaned off with lens paper. A baseplate was attached to the microscope (Inscopix) and positioned above the lens. The field of view (FoV) was visualized with nVista HD software (Inscopix) and the microscope carefully adjusted to be parallel to the lens surface. The microscope was lifted towards the best possible visualization of vasculature and putative neurons (approx. 200–250 µm above the focus of the lens surface). Before applying the dental cement, the microscope was raised another 50 µm. The baseplate was firmly attached to the previous implant with dental cement and a final layer of blackened cement was again applied to shield the lens from light. Once the cement hardened, we removed the microscope and attached a baseplate cover (Inscopix).

## Behavioral testing

Mice were tested in a series of behavioral paradigms while calcium imaging was performed. Videos were acquired with a CCD camera (OMRON SENTECH) at 30 frames/second. Prior to any experiment, mice were allowed to rest in a quiet holding area for at least 1 hr. Each day, mice would first undergo a 6 min recording in a familiar context (Makrolon cage with bedding, baseline recording). After 2 hr mice were tested in an open field arena (OF) Plexiglas box, 41 × 41 cm (Kinder Scientific) for 15 min (day –3), an elevated plus maze (gray taped Plexiglass, 16cm x 5 cm arms with two open arms and two arms with vertical walls 1 m above ground) for 6 min (day –2) and a contextual fear conditioning paradigm (day –1 to day 16). For the contextual fear paradigms, we used two different contexts. Context A consisted of a squared conditioning chamber (18 × 18 × 30 cm) with two metal and two clear walls and a stainless-steel grid floor (Coulbourn Instruments). Context C consisted of a round, white coated paper bucket with blue stripes (18 Ø x 30 cm). Each recording was performed for 6 min. Mice were pre-exposed to context A without a foot-shock on day-1 (preA). One day later, mice were re-introduced to context A and received three foot-shocks (2 s, 0.75mA) at 180 s, 240 s and 300 s (training day, day 0). The next day, mice were tested in context A and 2 hr later in context C (day 1). The order of context was counterbalance to avoid procedural learning. This was repeated 10 days (day 10) and 16 days after training in context A (day 16).

Mice with tetrode implants were recorded in a familiar context (Makrolon cage with bedding) 24 hr before and 15 min after receiving foot-shocks in context A (same parameters as above). Each recording session lasted 2 hours.

## Quantification of behavior

Behavioral videos were exported as AVI from Freezeframe (Actimetrics) and analyzed with Ethovision 15 (Noldus). Freezing behavior was quantified using the activity feature that is based on changes in pixel and therefore more robust against artefacts from the cable. Moving was measured as motion of the body center point.

## Calcium imaging acquisition

Mice were allowed to recover at least a week before being handled. Once mice were recovered, they were handled using the tunnel handling approach (see https://www.nc3rs.org.uk/mouse-handling-poster) and habituated to the microscope. All recordings were performed using nVista HD (Inscopix). Prior to recording sessions, mice were tested for sufficient calcium signal in the FoV, a reference image was taken, and the LED intensity tested to be optimal for recording. The LED intensity for each animal was maintained throughout the experiment. During the behavior paradigm, the microscope (nVista2, Inscopix) was attached to the baseplate and, when necessary, the FoV adjusted to match the reference image. Average number of recorded neurons per mouse and session were as follow: CA1: shNT mean:137, S.E.M. 15, shRNA mean:161, S.E.M. 20, ACC: shNT mean:89, S.E.M 21, shRNA mean: 68,

S.E.M 21. Videos were recorded at 20 frames/s and a resolution of 1440 × 1280 pixels. We used a TTL-pulse to align the onset of behavioral and calcium video recording.

### LFP eecordings

Simultaneous LFP recordings were performed using OpenEphys software and acquisition board (https://open-ephys.github.io). Signals were amplified with a 16 channel headstage that included an accelerometer (Intan, RHD 2132). During recordings, signals were sampled at 30,000 Hz and bandpass filtered between 0.1 and 6000 Hz. Only mice with stable LFP signals in both CA1 and ACC and low noise level were included in the study.

### Perfusion and histology

Mice were perfused with 4% PFA after completion of the behavioral paradigm. Brains were stored overnight in 4% PFA at 4 °C before being placed in 30% sucrose solution for 3 days at 4 °C. Brains were embedded in Optimal Cutting Temperature medium (OCT, Fisher HealthCare) and stored at –80 °C. Histological coronal sections (35 µm) were generated with a Leica cryostat (Leica) and stored in PBS with 0.01% sodium azide at 4 °C. Sections were immunohistologically stained as follow: Free floating sections were washed in PBS triton (0.3%) and unspecific binding sides blocked with blocking buffer (10% Normal Donkey Serum in PBS 0.3% Triton) for 2 hr at room temperature. Sections were further incubated with primary antibodies (as specified below) diluted in PBS 0.3% triton overnight at 4 °C. The next day, sections were washed three times (10 min) in PBS and incubated with secondary antibodies (as specified below) diluted in PBS for 2 hr at room temperature.

To validate the expression of the lenti-virus in all mice included in this study, we stained sections using antibodies against the fluochrome expressed by the lenti-virus (GFP or mCherry). A total of 4 mice across all groups were excluded from this study due to insufficient lenti-virus expression.

To quantify the effect of lenti-virus injection on inhibition, we used PV puncta as a histological marker as demonstrated previously (see Guo et al). See table for primary antibody details. We used alexa-fluor secondary antibodies (donkey, Jackson Immuno Research, 1:500 dilution) targeted against the host of the primary antibody.

| Target | Antibody (primary) | Fluor of 2nd AB | Purpose |
|---|---|---|---|
| GFP | Chicken anti-GFP (Aves, GFP-1020, 1:2000) | 488 | Label lenti-virus expression |
| mCherry | Rabbit anti-RFP (Rockland, 600-401-379, 1:200) | Cy3 | Label lenti-virus expression |
| Parvalbumin | Mouse anti-PV (EMD Millipore, MAB 1572, 1:2000) | Cy5 | Quantify PV puncta |

### PV quantification

We quantified the level of PV puncta in dorsal CA3, dorsal CA1 and ACC. For that, we used a confocal microscope (Leica TCS SP8) with 60 x oil objective to generate high-resolution images. Images were acquired using a sequence to reduce cross-talk. We imaged 18 ROIs per mouse (three brain sections, 3 ROIs per hemisection). We used a zoom of 4 x for CA3, and 2 x for CA1 and ACC. Images were blinded with ImageJ randomizer macro (macro by Tiago Ferreira) and scored manually. In CA1, we distinguished between superficial and deep layer by drawing a 20 µm deep square from either the upper (deep layer) or lower (superficial) border of dense DAPI signal.

### Calcium video processing

Videos were decompressed and preprocessed with Inscopix Data Processing Software (Inscopix) using the motion correction and temporal down sampling to 10 Hz. Preprocessed videos were exported as TIFF and further analyzed with MATLAB 2018a (Mathworks).

### Spatial and temporal calcium signal extraction

We used CNMF-E (*Zhou et al., 2018*) to extract spatial and temporal traces of individual neurons. We developed a MATLAB based code to further filter the extracted traces from CNMF-E to ensure, to the best of our ability in larger data sets, that our analysis is free from artefacts such as noise. We did so

by removing: (1) possible false positive neurons based on contour features, (2) possible false positive (noisy) calcium events, (3) neurons with low activity after removing noisy events, (4) low signal neurons with high overlapping neighbor neurons.

We removed elongated shapes using the polygeom function. If the inertia ratio was larger than 5, we rejected the neuron to exclude possible dendrites or blood vessels or visually distorted neurons. Furthermore, we removed neurons with rough contours that likely stemmed from background fluorescence using a simple linear classifier. The roughness, called compacity in this case, is calculated as:

$$C = \frac{P^2}{4\pi A}$$

where C is the compacity, P the perimeter and A the area of the neuron. The compacity threshold was empirically determined at 35. While CNMF-E includes a criteria to restrict overlap of neurons, we still found a small set of highly overlapping neurons in our extractions. To avoid a time-costly re-iteration of CNMF-E, we included this part of the CNMF-E approach into our pipeline. We identified candidate pairs with close proximity using an empirically tested threshold of 36 pixels between the center of each neuron. The overlap was then calculated as follow:

$$overlap = \frac{1}{2}\left( \frac{Area_{N_1 \cap N_2}}{Area_{N_1}} + \frac{Area_{N_1 \cap N_2}}{Area_{N_2}} \right)$$

where $Area_{N_1 \cap N_2}$ is the overlap area between the two neurons 1 and 2 and $Area_{N_1}$ and $Area_{N_2}$ the number of pixels of neurons 1 and 2, respectively. The area of neurons and the overlap area were calculated using the *polyshape* function. If neurons overlapped more than 50%, the neuron with lower PNR was removed.

To remove noisy calcium events, we used a 2-step approach. First, we extracted the timestamp of possible calcium events (event-candidates) using the *findpeak* function on the raw fluorescence traces and the denoised traces from CNMF-E (neuron.C_raw, neuron.C). A peak was defined by an amplitude of at least 3 SD from baseline noise, a minimum distance between two peaks of 15 frames (1.5 second) and a minimum duration of 3 frames (300ms). We further extracted 8 features of the event-candidates from both the raw and denoised fluorescence traces based on the *findpeak* function: Peak-to-noise ratio (PNR), Width-to-Height ratio, Peak prominence, Pearson correlation between peaks in the raw and denoised traces, and local PNR of the event candidate in the raw trace. For the local PNR, we calculated the standard deviation of the baseline in the vicinity of the peak rather than the overall recording baseline. We performed a PCA (*pca* function) on those eight parameters for dimension reduction to extract the components of these parameters that could be best used for classification. To reliable identify and remove calcium events that we considered noisy, we then trained a quadratic Bayesian classifier (*fitcdiscr*) on manually classified calcium peaks. After removing noise calcium events, neurons with low activity were removed using a linear classifier

$$N_{peaks} + 2 \times H_{max} - 3 > 0$$

where $N_{peaks}$ is the neuron's number of calcium events within one calcium recording and $H_{max}$ the relative height of the maximal calcium peak. We updated the CNMF-E output (neuron.A, neuron.C and neuron.C raw) to remove all neurons indicated by our pipeline and generated a file with calcium event times (sparse time series) for each neuron matching CNMF-E indexes. The updated data set was used for all further analysis presented in the figures.

## Correlated pairs

We calculated the correlation of calcium events between neurons using the xcorr function on time series data. We added a 200 ms frame around the calcium event (*Barbera et al., 2016*). We generated a distribution of correlation using temporally shuffled calcium event timeseries and calculated a threshold (mean + 2 SD) based on this distribution. Correlated pairs were defined as pairs with a correlation above this threshold. For analysis of foot-shock induced changes in cross-correlated pairs, we only included 30 s extracts of the preFS and postFS time series data (see *Figure 2*). To compare numbers of correlated pairs across sessions (e.g. between context A and context C), we normalized the number of correlated pairs to the number of correlated pairs found in the same day baseline ($CC_{pairs\ ctx\ A}$ / $CC_{pairs\ baseline}$).

## FSR neurons

To identify FSR neurons, we z-scored the calcium activity (calcium events) of each neuron to its time-binned (5 s) activity (calcium events) prior to the foot-shock (*Pan et al., 2020*). Neurons that increased their activity above 2SD during or after any foot-shock were classified as FSR neurons. To further quantify the foot-shock response we used 30 s timebin before and after the foot-shock (preFS vs postFS) and calculated the normalized activity before and after each foot-shock. We use the average of all three foot-shocks (preFS vs postFS) to quantify the overall effect of foot-shock training on those cells, which is why the average postFS response can be smaller than the initial threshold of 2SD [threshold to qualify as FSR]. To test for re-activated FSR within the population of correlated neurons during recall, we used the following criteria (1) the cell is matched across 2 days that is the cell was active during training and day one recall (in at least one context), (2) it was identified as FSR cell during training, (3) it is correlated with another cell during recall. This is independent of the correlated partner (other cell in pair does not have to fulfill these criteria).

## Identification of A&C neurons

We used CellReg (*Sheintuch et al., 2017*) to register neurons across different sessions (contexts and days). Spatial maps of neuronal contours were extracted by reshaping the spatial CNMF-E output (neuron.A) and down sampled 4 x for faster processing. We used two different approaches to calculate a threshold for same neurons and different neurons, one is the probabilistic approach based on the distribution of our data and the other is a fixed threshold (see *Sheintuch et al., 2017* for details). For all our recordings, spatial correlation was more reliable than centroid distance. We experienced that sessions with larger inter-session intervals (i.e. day 1 and day 16) were less accurate registered using the probabilistic approach than sessions that occurred closely in time. Therefore, we used the probabilistic approach only on sessions that were maximally 24 hr apart (training day and day 1; day 10; day16, data in main figures). To ensure that our findings were not driven by the cell registration approach, we also registered neurons using a fixed spatial correlation threshold of 0.85 as that was the average threshold we obtained in the probabilistic approach. This further allowed us to register neurons across all sessions (training day to day 16) as shown in the figure supplements. A&C neurons were defined as neurons that were active in both sessions (session A and session C) on the same day.

GitHub: https://github.com/HannahTwarkowski/DG_CA3_FFI_consolidation (*Twarkowski, 2021* copy archived at swh:1:rev:f7353d1b9385cda731a943a2415da69875b5d05a).

## Quantification of A&C neurons

The number of A&C neurons was expressed as normalized jaccard similarity index (*Ahmed et al., 2020*)

$$J(A, C) = |\frac{|A \cap C|}{|A \cup C|}$$

with A = neurons in session A, C = neurons in session C. The jaccard similarity index describes the number of A&C neurons relative to the total number of neurons active in A and C. The number of A&C neurons however can be biased by the total number of neurons active during one session. Given that then pool of neurons that we record from is limited, recordings with larger number of neurons are stochastically more likely to contain A&C neurons than recordings with smaller number of neurons. To correct for this bias, we generated a distribution of 1000 jaccard similarity indices by randomizing neuronal indexes while keeping the number of active neurons recorded per session. The jaccard index was then z-scored to that distribution. Furthermore, z-scored indices were normalized to their minimum and maximum to allow cross object comparison.

## Decoding context

We trained a support vector using the MATLAB statistics and machine learning toolbox. To avoid overfitting, we only included mice that had at least 50 A&C neurons per day. Instead of withholding testing data, we used 10fold cross-validation to train and test our data. The decoder was based on time binned (5 s) calcium event time series with averaged events per time bin of A&C neurons for both sessions. Thus, we obtained 72 vectors per context with the length of all A&C neuron reflecting each neuron's activity at a given time point. We used the decoder to test for differences between these

vectors that could predict the context. For all mice, we generated 1000 shuffled data set of randomized calcium event time (maintaining the same number of total calcium events/neuron and neurons) to train the decoder and test prediction certainty. All randomized data sets resulted in chance level of prediction.

## Ripple and spindle detection

Analysis of LFP signals was performed offline using Matlab 2020b (Mathworks). One channel for each region was selected and used for analysis. LFP signals were downsampled to 1250 Hz and analyzed using the FMA toolbox (*Zugaro, 2018*). Accelerometer data were used to determine states of immobility with the function *QuietPeriods*. LFP power was visualized using *MTspectrum* to control for possible contamination of the signal with electrical noise that would interfere with sharp-wave ripple detection. *BrainStates* function was used to determine sleep states using ACC and CA1 LFP as well as accelerometer data. Only slow-wave sleep episodes were further analyzed.

CA1 ripples were detected using *FindRipples* on bandpass filtered CA1 LFP traces (100–250 Hz). Signals that exceeded 3 SD at the beginning/end and 5 SD at the peak with a minimum duration of 50ms and a maximum duration of 100ms were classified as ripples. Detected ripple events that were less than 20ms apart were merged. Occurrence rate was calculated as the total number of ripples divided by the total time in SWS.

ACC spindles were detected using *FindSpindles* function on bandpass filtered ACC LFP traces (12–15 Hz). Signals that exceeded 2.5 SD for a minimum duration of 200ms and a maximum of 2000ms were classified as spindles. Occurrence rate was calculated as the total number of spindles divided by the total time in SWS.

Coupled Ripples were defined as ripples that occured within a 1 s window (±0.5 s) around the center of a spindle using the function *RippleStats* and an edited version of *RippleStats* to extract time stamps for onset, offset and center of the respective signal. Analysis was also performed using a smaller window of 0.5 s (±0.25 s) with similar results (data not shown).

## Statistics

Statistical tests were performed with GraphPad Prism v9 (Graphpad Software). We used nonparametric tests for sample sizes smaller than 5. For data sets with at least 5 mice, we used two-tailed students t-test (with Welch correction for unpaired data), one sample t-test (against 0, 1 or 50, see *Supplementary file 1* and figures for details) or two-way ANOVA with repeated measures and Sidak's multiple comparison test when appropriate. For all tests, significance was set at < 0.05. Individual data points reflect the average per mouse. Number of mice per group are indicated by n. All tests were performed using two-tailed settings. Details for all statistical analysis are provided in *Supplementary file 1*.

## Acknowledgements

We thank members of Sahay lab, Liron Sheintuch and Yaniv Ziv for comments on the manuscript. H.T. was supported by a German Research Foundation (DFG, TW 84/1-1) Postdoctoral fellowship and is recipient of a Harvard Brain Initiative Travel Grant. V.S was recipient of a Harvard Medical School Bertarelli Program in Translational Neuroscience and Neuroengineering Masters research fellowship. AS acknowledges support from US National Institutes of Health Biobehavioral Research Awards for Innovative New Scientists (BRAINS) 1-R01MH104175, NIH-NIA 1R01AG048908-01A1, NIH 1R01MH111729-01, The Simons Collaboration on Plasticity and the Aging Brain, the James and Audrey Foster MGH Research Scholar Award, Ellison Medical Foundation New Scholar in Aging, the Inscopix Decode Award, Ellison Family Philanthropic support, Blue Guitar Fund, Alzheimer's Association research grant and The Simons Collaboration on Plasticity and the Aging Brain.

## Additional information

### Competing interests

Amar Sahay: is a named inventor on a patent for targeting Ablim3 to improve memory (US Patent 10,287,580). The other authors declare that no competing interests exist.

## Funding

| Funder | Grant reference number | Author |
|---|---|---|
| DFG | German Research Foundation (DFG,TW 84/1-1) Postdoctoral fellowship | Hannah Twarkowski |
| NIH | NIH-NIA 1R01AG048908-01A1 | Amar Sahay |
| Simons Foundation | SCPAB | Amar Sahay |

The funders had no role in study design, data collection and interpretation, or the decision to submit the work for publication.

## Author contributions

Hannah Twarkowski, Data curation, Formal analysis, Funding acquisition, Investigation, Methodology, Software, Supervision, Validation, Visualization, Writing – original draft, Writing – review and editing; Victor Steininger, Data curation, Formal analysis, Investigation, Methodology, Software; Min Jae Kim, Data curation, Investigation, Methodology, Software; Amar Sahay, Conceptualization, Funding acquisition, Investigation, Project administration, Resources, Supervision, Validation, Writing – original draft, Writing – review and editing

## Author ORCIDs

Hannah Twarkowski ![ORCID] http://orcid.org/0000-0002-4349-8924
Amar Sahay ![ORCID] http://orcid.org/0000-0003-0677-1693

## Ethics

All animals were handled, and experiments were conducted in strict accordance with procedures approved by the Institutional Animal Care and Use Committee at the Massachusetts General Hospital in accordance with NIH guidelines (IACUC 2011N000084).

## Decision letter and Author response

Decision letter https://doi.org/10.7554/eLife.70586.sa1
Author response https://doi.org/10.7554/eLife.70586.sa2

---

# Additional files

## Supplementary files

• Transparent reporting form
• Supplementary file 1. Complete Statistics Table.

## Data availability

All data generated or analysed during this study are included in the manuscript and supporting files. GitHub link is provided. https://github.com/HannahTwarkowski/DG_CA3_FFI_consolidation copy archived at swh:1:rev:f7353d1b9385cda731a943a2415da69875b5d05a.

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
