## [Editor Report]

This paper will be of interest to scientists across systems neuroscience or to those interested in how one component of a neural circuit contributes to downstream functions longitudinally. The techniques used in this paper allowed the authors to characterize how increasing feed forward inhibition in the dentate gyrus-CA3 hippocampal circuit impacts the formation and maintenance of context-specific ensembles in CA1 and the anterior cingulate cortex without directly stimulating the circuit.

---

## [Decision Letter]

**Decision letter after peer review:**

Thank you for submitting your article "A dentate gyrus-CA3 inhibitory circuit promotes evolution of hippocampal-cortical ensembles during memory consolidation" for consideration by *eLife*. Your article has been reviewed by 3 peer reviewers, and the evaluation has been overseen by a Reviewing Editor and Laura Colgin as the Senior Editor. The following individuals involved in review of your submission have agreed to reveal their identity: Liset Menendez de la Prida (Reviewer #1).

Essential revisions:

The reviewers unanimously found this manuscript on feedforward inhibition of the CA3-DG circuit and memory processing to be of interest to the neuroscience community. However, all reviewers noted serious concerns that dampened their enthusiasm for supporting publication of the current version at *eLife*. Reviewers felt that the current experimental results and analyses did not fully support the main claims made in the manuscript. If the authors can address the main concerns listed below, we would be willing to consider a revised version of the manuscript.

1) All three reviewers asked for substantial additional analyses and additional control groups for proper comparisons. Please see the detailed suggested analyses in the individual reviews below.

2) Additional data is needed to support the authors' conclusion that there is a direct relationship between the manipulation of DG-CA3 FFI and the network activity changes in CA1 and ACC, as well as the behavioral improvement.

We know addressing these two concerns requires a substantial amount of additional work so if you do not think you will be able to fully address these concerns, you may want to consider submitting this work elsewhere.

*Reviewer #1 (Recommendations for the authors):*

1. For all statistical tests, the degree of freedom and factor values should be reported.

2. There are some differences between ACC and CA1 groups in terms of the order of viral manipulations. Why? Useful to discuss whether and how this can influence results.

3. I found the last section of results and associated figure panels very useful to summarize results. However, I feel they would read better at Discussion. Importantly, for explanatory sketches in Figure 2i and 4c,d, the authors should note their correlation analyses only inform about the % of correlated pairs, not about connectivity strengths. Hence thin and thicker lines are not reflecting data. Also, in Figure 2i and 4c, the basal condition may better reflect data in 1g regarding more % of active neurons in CA1 shRNA

4. Is there any difference between center/periphery exploratory preferences between groups or any other potential confounding factor which could explain activity differences better than the experimental manipulations?

5. The authors should declare the number of foot-shock responding (FSR) cells in each analysis block and test whether there are differences between groups. For some analyses, the number of cells could be too small so that they may require some specific randomization tests to reinforce conclusion (e.g., randomly picking of similar small number of not-FSR cells).

6. The authors refer to ACC-CA1 correlations or CA1-ACC networks several times but this is not formally examined (ACC and CA1 imaging is independent therefore there are no formal cross-region comparisons in this paper).

7. The authors refer to effects of increased SWR-spindle coupling between CA1-ACC in Results section, which reads rather speculative.

8. Regarding conclusions and physiological relevance, the authors may need to discuss why enhanced feedforward inhibition at DG-CA3 synapses is not naturally established given the beneficial effect in context discrimination.

*Reviewer #2 (Recommendations for the authors):*

1) Throughout the manuscript, the authors claim that there is a significant increase in context A specific correlated pairs in CA1 when FFI was increased (Figure 3d, bottom panel), however this is not supported by the statistical data. Performing one-sample t-tests on each group is not statistically sound, and the authors report 0.055 as a seemingly significant effect (despite explicitly setting 0.05 as their α) and yet do not report p-values for other effects that seem close to significant by eye. This general approach does not seem valid. Furthermore, the broader interpretations of this paper (e.g. the teacher-like function of CA1) rely on this point being significant, so should be corrected to reflect that this is a trend rather than a statistically significant effect or additional experiments should be added to increase the power of the comparison.

2) Relatedly, there appears to be more variability in the shRNA group compared to what is seen in the controls, especially in measurements from CA1 (e.g. Figure 3d, 3k). Is it possible that this variability might be explained by variable levels of viral transfection? What steps were taken to ensure viral expression was comparable among all animals included in the experiments?

3) The claims that shRNA increases CA1 activity and correlated firing relative to ACC is not justified (Figure 1g,h). There is no rationale for this specific comparison and the entire rest of the paper compares the shNT and shRNA groups. There needs to be a strong rationale for this analysis or the comparisons should be between experimental groups, which the authors report as not significant for activity.

4) When comparing FSR cells, the increase in correlated pairs likely reflects higher activity since the cells are selected by having higher activity post-shock. The authors acknowledge this on p4: "Shuffled data indicate that the increase in correlated pairs was mostly due to increased number of calcium events." Yet this acknowledgement is not considered when the authors draw interpretations from their data and no data is presented showing how activity levels influence the correlated pairs. Any conclusions based on correlated pairs need to clarify how activity levels contributed to the synchronized firing.

5) The lack of a non-shock control group makes it difficult to interpret how selecting for foot shock responsive cells is influencing the data. Perhaps the authors could use baseline data to do a similar analysis to see how much selection bias is driving the findings of Figure 2.

6) It is unclear why calculations of correlated pairs are normalized to baseline in Figure 3, but not in Figure 2. In Figure 2, the primary finding that shRNA group increases the number of correlated pairs would seemingly go away if normalized to preFS levels (presumably similar to baseline). This variability makes it difficult to interpret the normalized values since there is clearly substantial variability in the baseline numbers. Perhaps reporting raw correlated pair numbers or adding new subjects would be more clear.

7) Figures 2i and 4c-d show schematics summarizing the data, but don't seem to represent the data related to correlation accurately. For example, in Figure 2h the authors show that there is an increase in the percentage of neuronal pairs that are co-active in CA1 in both the shNT and shRNA groups, and that this is increase in the number of co-active pairs is more extreme in the shRNA group (Supplementary Figure 1b). The schematic, however, implies that the strength of the correlation between co-active pairs is increased, which does not appear to have been tested.

8) Page 5 173 – "Using CellReg (Sheintuch et al., 2017) to register neurons across sessions, we found no reactivation of FSR neurons among the correlated pairs of neurons identified in CA1 in the recall session [shNT: mean: 0.0 S.E.M. 0; shRNA: mean 0.0, S.E.M, 0] (data not shown)." Does this mean that to reach this criterion both of the correlated cells would have to be matched across sessions and then again be correlated during recall? Please clarify this statement.

9) Figure 2f and 3n show pie charts without individual animal information. These are not very interpretable and should be replaced by graphs that showed animal by animal variability.

10) Page 5 – "We controlled for the possible effect of locomotion on neuronal activity by quantifying the time mice moved in each context and found no correlation between the number of correlated neuronal pairs and the time mice moved in the respective context [Pearson's R2, ACC context A, 0.19, ACC context C: 0.00, CA1 context A: 0.09, CA1 context C: 0.04, n=5 mice per group] (Supplementary Figure 2a-d)." Please add p-values as R2 values are a bit ambiguous.

11) The authors should consider that the increased activity in CA1 A&C neurons in context C may be driven by locomotion rather than novelty (Figure 3f and Supplemental Figure 2d).

*Reviewer #3 (Recommendations for the authors):*

1) To follow up on my comment 1), I think it would be useful to quantify the impact of the manipulation on CA3 activity in vivo. This can be done using electrophysiology or imaging. If this is beyond the scope of this study, an indirect read-out of CA3 activity could be obtained through a sharp wave ripple analysis in CA1, which would provide an independent indicator of how intrahippocampal information processing in vivo is changed by the manipulation.

2) It would be helpful to illustrate at the beginning of the paper how a calcium event is defined. For example, the data in figure 1g appear to show that the activity patterns in CA1 have been changed when increasing DG-CA3 FFI. The frequency is increased, but there is also an increase in "double-peak transients". How exactly are these handled by the analysis? Along the same lines, it would be helpful to clarify the y axis label 'event rate [SD]'? The term rate implies frequency, but if I understand it correctly, the analysis mixes amplitude and frequency, correct? Including some raw data would, I think, help understand how the analysis impacts the findings of the paper.

3) The graphics in Figure 1i and Figure 4c are misleading. There is no indication in the data about the strength of the correlation; the data presented only show that the fraction of correlated FSR neuron pairs is increased after footshock, not that there is a change in the correlation strength. If I understand Figure 2h correctly, there seems to be roughly a doubling in the fraction of correlated pairs in all groups, except for the shNT/ACC group showing a 4-5 times increase in the fraction of correlated pairs.

---

## [Author Response]

Essential revisions:The reviewers unanimously found this manuscript on feedforward inhibition of the CA3-DG circuit and memory processing to be of interest to the neuroscience community. However, all reviewers noted serious concerns that dampened their enthusiasm for supporting publication of the current version at eLife. Reviewers felt that the current experimental results and analyses did not fully support the main claims made in the manuscript. If the authors can address the main concerns listed below, we would be willing to consider a revised version of the manuscript.1) All three reviewers asked for substantial additional analyses and additional control groups for proper comparisons. Please see the detailed suggested analyses in the individual reviews below.

We performed additional analyses of our data to test for variables other than our virus manipulation that may explain our findings. None of these additional analyses provided sufficient evidence for an alternative variable (see Author response image 1). In addition, we performed new analysis (included in Figure 4 in the revised manuscript) that strengthens evidence for our original conclusion.

**Author response image 1. sa2fig1:** Additional analyses of calcium imaging and behavioral data as suggested by the reviewers. Figure references indicate the corresponding figure in the main, revised manuscript. (a-d) Analysis of behavioral data during training (a, time freezing) or recall (b-d) to test for difference between control groups (ACC shNT vs CA1 shNT). No difference between groups was found in any analysis [2-Way repeated measures ANOVA; (a) time x brain region effect, p=0.995, main brain region effect, p=0.109, main time effect: p=0.002, n=5; (b) time x brain region effect, p=0.647, main brain region effect, p=0.868, main time effect: p=0.019, n=5; (c) time x brain region effect, p=0.978, main brain region effect, p=0.443 , main time effect: 0.443 , n=5; (d) time x brain region effect, p=0.849, main brain region effect, p=0.969, main time effect: p=0.68; n=5;]. (e-f) Moving index during recall day 1 in one second time bins. No group difference was detected [2-Way repeated measures ANOVA; (e) time x virus effect, p=0.011, main virus region effect, p=0.749, main time effect: p=0.372, n=5; Sidak’s posthoc, N.S; (f) time x virus region effect, p=0.435, main virus effect, p=0.695, main time effect: 0.16, n=5]. (g) Correlation of decoder accuracy and behavioral discrimination ratio during both recall sessions [Pearson’s correlation, shNT: R2=0.579, p=0.028, n=8; shRNA: R2=0.577, p=0.029, n=8]. (h) Absolute numbers of FSR neurons [Unpaired t-test with Welch’s correction, ACC, p=0.705, n=5; CA1, p=0.393, n=5]. (i-j) Analysis of randomly picked nonFSR neurons for change in event rate (i) and correlated pairs (j) upon foot-shocks. In ACC, no difference in either measure was found [Paired t-test, event rate (i), shNT, p=0.123, n=5; shRNA, p=0.924, n=5; correlated pairs (j): shNT, p=0.052, n=5; shRNA, p=0.112, n=5]. In CA1 neurons showed a significant decrease in event rate in both groups [Paired t-test, event rate (i), shNT, p=0.034, n=5; shRNA, p=0.033, n=5] and a decrease in correlated pairs in the shNT group [Paired t-test, correlated pairs (j), shNT, p=0.044, n=5; shRNA, p=0.055, n=5]. (k) Number of shamFSR neurons found in baseline recording on the training day. shamFSRs were identified using the same approach as training day (activity change around foot-shock). Number of shamFSRs did not differ between treatment groups. [Unpaired t-test with Welch’s correction, ACC, p=0.702, n=5; CA1, p=0.09, n=5]. (l) Comparison of numbers of shamFSR neurons during baseline and FSR neurons during training. In ACC, shRNA mice show significantly more FSR neurons compared to shamFSR neurons [paired t-test with Welch’s correction, shNT, p=0.473, n=5; shRNA, p=0.021, n=5]. (m-n) Analysis of shamFSR neurons for change in event rate (m) and correlated pairs (n) upon time points similar to foot-shock time points. No change in event rate or correlated pairs for detected in shamFSR neurons in any group [Paired t-test, event rate (m), ACC, shNT, p=0.089, p=5, shRNA, p=0.075, n=5; CA1, shNT, p=0.118, n=5; shRNA, p=0.122, n=5; correlated pairs (n), CA1, shNT, p=0.063, n=5, shRNA, p=0.38, n=5; CA1, shNT, p=0.088, n=5, shRNA, p=0.086, n=5]. (o) Correlation of activity of correlated pairs and number of correlated pairs during recall (left) and comparison of activity between correlated and uncorrelated neurons (middle-right) in CA1. A negative correlation was found between the number of correlated pairs and activity of correlated neurons in shRNA mice. [Pearson’s correlation, shNT, R2=0.027, p=0.493, n=20, shRNA, R2=0.236, p=0.03, n=20]. No significant difference was found between activity in correlated and uncorrelated neurons [Paired t-test, shNT, p=0.06, n=20, shRNA, p=0.085, n=20]. (p) Number of correlated pairs during baseline recording on training day [Unpaired t-test with Welch’s correction, ACC, p=0.774, n=5; CA1, p=0.108, n=5]. (q) Correlation between event rate of A&C neurons in CA1 and time moving in context C during recall sessions day1 and day 16. No correlation is found in either group [Pearson’s correlation, shNT, R2=0.113, p=0.344, n=10, shRNA, R2=0.043, p=0.563, n=10].

2) Additional data is needed to support the authors' conclusion that there is a direct relationship between the manipulation of DG-CA3 FFI and the network activity changes in CA1 and ACC, as well as the behavioral improvement.

We performed additional experiments as suggested by the reviewers to test our hypothesis of a direct relationship between DG-CA3 FFI and CA1-ACC interaction. To directly test hippocampal cortical communication following viral mediated enhancement of FFI in DG-CA3, we performed simultaneous LFP recordings in CA1 and ACC and extracted sharp-wave ripples in CA1 and spindles in ACC. We found increased ripple-spindle coupling in mice in which we increased FFI in DG-CA3.

Reviewer #1 (Recommendations for the authors):1. For all statistical tests, the degree of freedom and factor values should be reported.

We included the degree of freedom and factors in Supplementary File 1. For readability, we decided to not include them in the main text or figure legend.

2. There are some differences between ACC and CA1 groups in terms of the order of viral manipulations. Why? Useful to discuss whether and how this can influence results.

In pilot experiments, we found that simultaneous injection of lenti-virus in the dDG and AAV in the dCA1 did not yield high levels of infection as ascertained by analysis of expression. Therefore, we separated the surgeries by 3 days to ensure proper infection that was comparable to mice in the ACC cohort. In addition, we used an angled injection into dDG to reduce impact on dCA1. Interestingly, the order used in our study (AAV first, then lenti virus) was found to be most efficient for high levels of expression of both AAV and lenti payloads. In line with animal welfare, we decided to maintain the original (two injections in one surgery) protocol for the ACC cohort. The difference in protocol could impact our data in regard to the duration of manipulation (lenti-virus) but we think this is highly unlikely.

3. I found the last section of results and associated figure panels very useful to summarize results. However, I feel they would read better at Discussion. Importantly, for explanatory sketches in Figure 2i and 4c,d, the authors should note their correlation analyses only inform about the % of correlated pairs, not about connectivity strengths. Hence thin and thicker lines are not reflecting data. Also, in Figure 2i and 4c, the basal condition may better reflect data in 1g regarding more % of active neurons in CA1 shRNA

Thank you. We edited the sketch to (1) better reflect the number of correlated pairs instead of the strength and (2) the differences in preFS conditions (Figure 2i and new Figure 6a).

4. Is there any difference between center/periphery exploratory preferences between groups or any other potential confounding factor which could explain activity differences better than the experimental manipulations?

It is true that behavior is very complex and our readouts of freezing or movement only capture some aspects of it. However, we could not identify another factor that explained the differences better than our manipulation as evidenced in the lack of correlation between cross-correlated pairs and time moving (suppl Figure 2d). It might be interesting to look into neural activity as the mice explore center vs periphery and potentially resolve competing goals (avoidance vs approach etc), but our recording chambers are too small (context A: 18 x 18 cm, context C 18 cm diameter) to reliable distinguish subfields and the position of the mouse within those.

5. The authors should declare the number of foot-shock responding (FSR) cells in each analysis block and test whether there are differences between groups. For some analyses, the number of cells could be too small so that they may require some specific randomization tests to reinforce conclusion (e.g., randomly picking of similar small number of not-FSR cells).

Thank you for this great suggestion. As shown in Author response image 1, the absolute number of FSR neurons did not differ between groups or brain regions. Once a neuron was identified as FSR, the neuronal activity of all 3 time blocks (around all 3 foot shocks) was included in the analysis hence the number of FSR is the same for all blocks.

We strongly agree with you that the small number of FSR neurons may require a different test to reinforce our conclusions and appreciate the reviewer’s suggestions how to address that. We performed two new analyses on our data.

First, as suggested here, we randomly picked the same number of neurons (FSR neurons) from the pool of non-FSR neurons (nFRS neurons) and repeated the same analysis regarding normalized event rate and correlated pairs on those non-FSR neurons. We found no increase in event rate or correlated pairs in any group (see Author response image 1) but rather a reduction in event rate in CA1 (Two-tailed paired t-test, shNT p=0.034, n=5, shRNA p=0.033, n=5) and no change in ACC in the small subset of nFSR neurons (Author response image 1) (see Figure 2 e, h in manuscript for comparison).

6. The authors refer to ACC-CA1 correlations or CA1-ACC networks several times but this is not formally examined (ACC and CA1 imaging is independent therefore there are no formal cross-region comparisons in this paper).

This is a great point and was raised by all reviewers. We acknowledge the weakness of this comparison, apologize for this misstep in our analysis and have accordingly, removed this dataset from our manuscript. Instead, we performed new experiments using in vivo electrophysiology to allow for cross-region comparison of LFPs in CA1 and ACC within the same animal. We removed data from Figure 1 e-i and added new, simultaneous electrophysiological LFP recordings (Figure 5 and Figure 5 —figure supplement 1 in revised manuscript).

7. The authors refer to effects of increased SWR-spindle coupling between CA1-ACC in Results section, which reads rather speculative.

We added new simultaneous LFP recordings to address this question in Figure 5. We found an increased number of CA1 ripples that are coupled with ACC spindles (“coupled ripples”) in shRNA mice compared to control mice prior to a learning event (Figure 5c, two-tailed unpaired student’s t-test with Welch’s correction, p=0.0499, n=5) with no difference in time spend in slow-wave sleep (SWS) (Figure 5 —figure supplement 1a) or total numbers of spindles or ripples (Figure 5 —figure supplement 1b-c). Control mice show a learning-dependent increase in coupled ripples (Figure 5f, two-tailed paired student’s t-test, p=0.019, n=5) to a similar level as seen in shRNA mice prior to learning. No further increase is seen in shRNA mice indicating a saturation of circuit changes that cannot be further amplified following learning.

8. Regarding conclusions and physiological relevance, the authors may need to discuss why enhanced feedforward inhibition at DG-CA3 synapses is not naturally established given the beneficial effect in context discrimination.

We apologize that we did not make that aspect more clear in our discussion. We edited the abstract, introduction and discussion to convey that enhanced feedforward inhibition at the DG-CA3 synapses is naturally temporarily established following a spatial learning event (see Ruediger et al. 2011, Ruediger 2012, Guo et al., 2018). (LL 65, LL 365)

Reviewer #2 (Recommendations for the authors):(1) Throughout the manuscript, the authors claim that there is a significant increase in context A specific correlated pairs in CA1 when FFI was increased (Figure 3d, bottom panel), however this is not supported by the statistical data. Performing one-sample t-tests on each group is not statistically sound, and the authors report 0.055 as a seemingly significant effect (despite explicitly setting 0.05 as their α) and yet do not report p-values for other effects that seem close to significant by eye. This general approach does not seem valid. Furthermore, the broader interpretations of this paper (e.g. the teacher-like function of CA1) rely on this point being significant, so should be corrected to reflect that this is a trend rather than a statistically significant effect or additional experiments should be added to increase the power of the comparison.

Thank you for raising this valid point. Our aim was to describe the data in reference to the baseline in order to test for neuronal ensembles that may form in response to exposure to a context. To do so, without presenting complicated graphs and posthoc comparisons, we decided to present the data in the current format. Importantly, we avoided multiple comparisons or to draw conclusions across groups and thus did not affect our α level. Based on your remarks, we decided to present the data in a different format. We now show data per context and perform student’s t-test to address across group differences. See edits in Figure 3d,k. However, we still think it is important to test within group to address the question of context-related correlated pairs. Thus, we perform a one-sample t-test within each group and strictly restrict conclusion drawn from that to the tested group as done in similar prior publications. Please refer: (Cai et al., 2016; Fernández-Ruiz et al., 2017; Maviel et al., 2004; Schuette et al., 2020; Stark et al., 2014).

Cai, D. J., Aharoni, D., Shuman, T., Shobe, J., Biane, J., Song, W.,... Silva, A. J. (2016). A shared neural ensemble links distinct contextual memories encoded close in time. *Nature*, *534*(7605), 115-118.

Fernández-Ruiz, A., Oliva, A., Nagy, G. A., Maurer, A. P., Berényi, A., & Buzsáki, G. (2017). Entorhinal-CA3 Dual-Input Control of Spike Timing in the Hippocampus by Theta-Γ Coupling. *Neuron*, *93*(5), 1213-1226.e1215.

Maviel, T., Durkin, T. P., Menzaghi, F., & Bontempi, B. (2004). Sites of neocortical reorganization critical for remote spatial memory [Research Support, Non-U.S. Gov't]. *Science*, *305*(5680), 96-99.

Schuette, P. J., Reis, F. M. C. V., Maesta-Pereira, S., Chakerian, M., Torossian, A., Blair, G. J., Adhikari, A. (2020). Long-Term Characterization of Hippocampal Remapping during Contextual Fear Acquisition and Extinction. *J Neurosci*, *40*(43), 8329-8342.

Stark, E., Roux, L., Eichler, R., Senzai, Y., Royer, S., & Buzsaki, G. (2014). Pyramidal cell-interneuron interactions underlie hippocampal ripple oscillations. *Neuron*, *83*(2), 467-480.

We reported the exact p-value both in the figure and the legend while phrasing that it is significant (p<0.05) in the main text and apologize that our text conveyed otherwise. As described above, we edited the figure, analysis and text to address this issue. We are not aware of any graphs that indicate a significant p-value where we failed to report results. All tests can be found in the supplementary statistic table.

We agree that we need to state it correctly, yet we strongly disagree that it weakens our broader conclusion. First, α levels in biological experiments have been used broadly as 0.05 without proper proof of biological relevance (compared to 0.06 or 0.04). Second, the interpretation that CA1 may instruct ACC is guided by our finding that the strength of neuronal representation (CC pairs, A&C neurons within CCpairs) in shRNA mice at remote time points is accompanied by increased precision in ACC (A&C neurons) (Figure 3k,m,n). And our new electrophysiological recordings that demonstrate increase CA1-ACC synchrony in animals in which we increase FFI in DG-CA3.

2) Relatedly, there appears to be more variability in the shRNA group compared to what is seen in the controls, especially in measurements from CA1 (e.g. Figure 3d, 3k). Is it possible that this variability might be explained by variable levels of viral transfection? What steps were taken to ensure viral expression was comparable among all animals included in the experiments?

First, based on past experience using shRNA virus to downregulate Ablim3 and increase FFI in DG-CA3 (Guo et al., 2018) we only included mice with sufficient infection in the dDG. We now provide a figure with all animals included in the study (see Author response image 2). All mice show sufficient expression of the lenti-virus construct. Furthermore, the functional impact of Ablim3 shRNA expressed by the lenti-virus is naturally limited to DG cells given that ablim3 is solely expressed in this cell population (https://hipposeq.janelia.org/full/t1/E18E148C-3805-11EC-A082-ECE49DC49958/) and not in adjacent CA3c or CA3ab or CA1. Second, Terminals of DG granule cell axons (mossy fiber, MF) form synapses onto CA3 pyramidal neurons with low convergence while mossy fiber filopodia innervate parvalbumin positive (PV) interneurons with high convergence. Thus, the level of infection may not have a linear relation to the level of FFI provided within the DG-CA3 network.

**Author response image 2. sa2fig2:** Viral expression of lenti-shNT/ shRNA construct in the dorsal DG in all mice included in this study. Representative images are shown for each mouse.

(3) The claims that shRNA increases CA1 activity and correlated firing relative to ACC is not justified (Figure 1g,h). There is no rationale for this specific comparison and the entire rest of the paper compares the shNT and shRNA groups. There needs to be a strong rationale for this analysis or the comparisons should be between experimental groups, which the authors report as not significant for activity.

This is a great point and was raised by all reviewers. We acknowledge the weakness of this comparison, apologize for this misstep in our analysis and have accordingly, removed this dataset from our manuscript. Instead, we performed new experiments using in vivo electrophysiology to allow for cross-region comparison of LFPs in CA1 and ACC within the same animal. We removed data from Figure 1 e-i and added new, simultaneous electrophysiological LFP recordings (Figure 5 and Figure 5 —figure supplement 1 in revised manuscript).

(4) When comparing FSR cells, the increase in correlated pairs likely reflects higher activity since the cells are selected by having higher activity post-shock. The authors acknowledge this on p4: "Shuffled data indicate that the increase in correlated pairs was mostly due to increased number of calcium events." Yet this acknowledgement is not considered when the authors draw interpretations from their data and no data is presented showing how activity levels influence the correlated pairs. Any conclusions based on correlated pairs need to clarify how activity levels contributed to the synchronized firing.

We partly addressed that by showing the average firing rate during the recall sessions in Figure 3 —figure supplement 1 j,l and see that in CA1 calcium event rate is actually lower in context A compared to context C while we find more correlated pairs in context A (see Figure 3 d,k). However, averaging over mice may indeed occlude a potential link between neuronal activity and correlated pairs. We provide additional analysis showing the number of correlated pairs plotted against neuronal activity of correlated neurons (averaged over time, as correlated pairs are also calculated over the entire time) (Author response image 1). We find no positive correlation between these two parameters indicating that it is not a general increase in neuronal activity that leads to the findings of correlated pairs in context A. We rather find a negative correlation in shRNA mice (Pearson’s correlation, shNT: R^2^=0.027, p=0.493; shRNA: R^2^=0.236, p=0.03). Comparison between neuronal activity shows a tendency towards higher activity in non-correlated neurons in both groups (Author response image 1, middle-right, Paired t-test , shNT, p=0.06, n=20, shRNA, p=0.085, n=20). Thus, it seems that neuronal activity does not drive the increase in correlated pairs found during recall.

We also addressed that in the main text (LL 176).

5) The lack of a non-shock control group makes it difficult to interpret how selecting for foot shock responsive cells is influencing the data. Perhaps the authors could use baseline data to do a similar analysis to see how much selection bias is driving the findings of Figure 2.

We strongly agree that the small number of FSR neurons may require a different test to reinforce our conclusions and appreciate the reviewers’ suggestions how to address that. We performed two new analyses on our data.

First, as suggested here, we randomly picked the same number of neurons (FSR neurons) from the pool of non-FSR neurons (nFRS neurons) and repeated the same analysis regarding normalized event rate and correlated pairs on those non-FSR neurons. We found no increase in event rate or correlated pairs in any group (see Author response image 1) but rather a reduction in event rate in CA1 (Two-tailed paired Student’s t-test, shNT p=0.034, n=5, shRNA p=0.033, n=5) and no change in ACC in the small subset of nFSR neurons (Author response image 1).

Second, we performed the original analysis to find FSR neurons (“shamFSR”) on the same day Baseline recording as a control recording that did not include a foot-shock. Surprisingly, we found a similar number of neurons that were characterized as shamFSR compared to FSR neurons found in the training session (Author response image 1, see Figure 2 —figure supplement 1a in manuscript for comparison) in CA1 as well as in the ACC shNT mice. Thus, indicating that neuronal activity of a subset of neurons varies within sessions independent of the foot-shocks in those groups. Originally, we reported that shRNA manipulation increases the number of FSR neurons in ACC. To test if this is still valid with the new finding of shamFSR neurons, we compared the percentage of shamFSR neurons (from baseline recording) with the originally reported percentage of FSR neurons in ACC mice. We found that shRNA injected mice show a significantly higher number of FSR neurons compared to shamFSR neurons supporting our original findings (Author response image 1, paired t-test, shNT: N.S, n=5; shRNA p=0.0214; n=5). We found no difference in neuronal activity or number of correlated pairs in shamFSR neurons (Author response image 1) indicating that the increase in activity and correlated pairs reported in the original manuscript is specific to foot-shock learning.

6) It is unclear why calculations of correlated pairs are normalized to baseline in Figure 3, but not in Figure 2. In Figure 2, the primary finding that shRNA group increases the number of correlated pairs would seemingly go away if normalized to preFS levels (presumably similar to baseline). This variability makes it difficult to interpret the normalized values since there is clearly substantial variability in the baseline numbers. Perhaps reporting raw correlated pair numbers or adding new subjects would be more clear.

Our longitudinal study design includes many different variables that may affect our recordings such as previous experience (e.g. previous recording). We tried to use normalization to account for cross-animal variability and effects of such previous experiences and therefore chose the temporally most proximal reference point to normalize to. In Figure 2 and 3 we asked 2 different questions and therefore used other approaches. In the training session (Figure 2), we wanted to test the effect of a foot-shock while mice are in context A. Therefore, our reference is context A prior to the foot-shock within the same session. In the recall sessions, we wanted to test the effect of context, so our reference is the familiar context of the same day (baseline session). In Author response image 1 we provide the percentage of correlated pairs during the baseline recording on the training day. We don’t find a difference between groups in either ACC or CA1 that could drive our findings in Figure 2. Moreover, in Figure 2 —figure supplement 1b we also show the change in correlated pairs, which is a form of normalization to preFS level as suggested here, and find that indeed shRNA mice show a significantly higher change in correlated pairs compared to shNT injected mice.

7) Figures 2i and 4c-d show schematics summarizing the data, but don't seem to represent the data related to correlation accurately. For example, in Figure 2h the authors show that there is an increase in the percentage of neuronal pairs that are co-active in CA1 in both the shNT and shRNA groups, and that this is increase in the number of co-active pairs is more extreme in the shRNA group (Supplementary Figure 1b). The schematic, however, implies that the strength of the correlation between co-active pairs is increased, which does not appear to have been tested.

Thank you for raising this point. We edited the sketch to (1) better reflect the number of correlated pairs instead of the strength and (2) the differences in preFS conditions (Figure 2i and new Figure 6a).

8) Page 5 173 – "Using CellReg (Sheintuch et al., 2017) to register neurons across sessions, we found no reactivation of FSR neurons among the correlated pairs of neurons identified in CA1 in the recall session [shNT: mean: 0.0 S.E.M. 0; shRNA: mean 0.0, S.E.M, 0] (data not shown)." Does this mean that to reach this criterion both of the correlated cells would have to be matched across sessions and then again be correlated during recall? Please clarify this statement.

We apologize if that was not clear. Only 1 cell of the correlated pair is sufficient. In detail, the criteria for this analysis are (1) the cell is matched across 2 days meaning that it was active during training and day 1 recall (in at least one context), (2) it was identified as FSR cell during training, (3) it is correlated with another cell during recall. This is independent of the correlated partner (other cell in pair does not have to fulfill these criteria). This is now included in the method section for FSR neurons.

9) Figure 2f and 3n show pie charts without individual animal information. These are not very interpretable and should be replaced by graphs that showed animal by animal variability.

These data can be found in the bar graph (Figure 2 —figure supplement 1a (for Figure 2f) and Figure 3 —figure supplement 1 e-h (for Figure 3n)). We tried to reduce the number of bar graphs to make the figures easier to read. We added the mean and SEM for each chart in the legend.

10) Page 5 – "We controlled for the possible effect of locomotion on neuronal activity by quantifying the time mice moved in each context and found no correlation between the number of correlated neuronal pairs and the time mice moved in the respective context [Pearson's R2, ACC context A, 0.19, ACC context C: 0.00, CA1 context A: 0.09, CA1 context C: 0.04, n=5 mice per group] (Supplementary Figure 2a-d)." Please add p-values as R2 values are a bit ambiguous.

We provided those in the statistical table but agree that they need to be stated in the text. We added these values in Figure 3 —figure supplement 1d, Figure 3 —figure supplement 2b and in the text.

11) The authors should consider that the increased activity in CA1 A&C neurons in context C may be driven by locomotion rather than novelty (Figure 3f and Supplemental Figure 2d).

Thank you for raising this point. In Author response image 1 we provide additional analysis in CA1 mice. We show the normalized event rate in A&C neurons (from Figure 3f,l) against the percentage of time mice spent moving in context C (Figure 3 —figure supplement 1d and newly included data from day 16) (Pearson’s correlation, shNT, R^2^=0.113, p=0.344; n=10; shRNA, R^2^=0.208, p=0.563, n=10).

Reviewer #3 (Recommendations for the authors):1) To follow up on my comment 1), I think it would be useful to quantify the impact of the manipulation on CA3 activity in vivo. This can be done using electrophysiology or imaging. If this is beyond the scope of this study, an indirect read-out of CA3 activity could be obtained through a sharp wave ripple analysis in CA1, which would provide an independent indicator of how intrahippocampal information processing in vivo is changed by the manipulation.

This is a great point and was raised by all reviewers. We acknowledge the weakness of this comparison, apologize for this misstep in our analysis and have accordingly, removed this dataset from our manuscript. Instead, we performed new experiments using in vivo electrophysiology to allow for cross-region comparison of LFPs in CA1 and ACC within the same animal. We removed data from Figure 1 e-i and added new, simultaneous electrophysiological LFP recordings (Figure 5 and Figure 5 —figure supplement 1 in revised manuscript).

We found an increased number of CA1 ripples that are coupled with ACC spindles (“coupled ripples”) in shRNA mice compared to control mice prior to a learning event (Figure 5c, two-tailed unpaired student’s t-test with Welch’s correction, p=0.0499, n=5) with no difference in time spend in slow-wave sleep (SWS) (Figure 5 —figure supplement 1a) or total numbers of spindles or ripples (Figure 5 —figure supplement 1b-c). Control mice show a learning-dependent increase in coupled ripples (Figure 5f, two-tailed paired student’s t-test, p=0.019, n=5) to a similar level as seen in shRNA mice prior to learning. No further increase is seen in shRNA mice indicating a saturation of circuit changes that cannot be further amplified following learning.

2) It would be helpful to illustrate at the beginning of the paper how a calcium event is defined. For example, the data in figure 1g appear to show that the activity patterns in CA1 have been changed when increasing DG-CA3 FFI. The frequency is increased, but there is also an increase in "double-peak transients". How exactly are these handled by the analysis? Along the same lines, it would be helpful to clarify the y axis label 'event rate [SD]'? The term rate implies frequency, but if I understand it correctly, the analysis mixes amplitude and frequency, correct? Including some raw data would, I think, help understand how the analysis impacts the findings of the paper.

We agree that understanding the extraction of events is important aspect of calcium imaging. We included a trace of raw calcium dynamics and marked detected events to visualize our approach (Figure 1e, lower trace). Events were detected based on the following restrictions: minimum peak height of 3 times standard deviation of the baseline noise, minimum distance between two peaks of 15 frames (1.5 second), minimum peak width of 3 frames (300 ms). Thus, “double-peak transients” were included if they fulfilled these criteria.

The event rate is indeed the numbers of events/time bin and therefore implies frequency. The amplitude was only considered to detect an event. Analysis of events (i.e. rate or correlation) was based on binary time-series data indicating the timepoint of an event.

3) The graphics in Figure 1i and Figure 4c are misleading. There is no indication in the data about the strength of the correlation; the data presented only show that the fraction of correlated FSR neuron pairs is increased after footshock, not that there is a change in the correlation strength. If I understand Figure 2h correctly, there seems to be roughly a doubling in the fraction of correlated pairs in all groups, except for the shNT/ACC group showing a 4-5 times increase in the fraction of correlated pairs.

We edited the sketch to (1) reflect better the number of correlated pairs instead of the strength and (2) the differences in preFS conditions (Figure 2i and new Figure 6a).

References

Cai, D. J., Aharoni, D., Shuman, T., Shobe, J., Biane, J., Song, W.,... Silva, A. J. (2016). A shared neural ensemble links distinct contextual memories encoded close in time. *Nature*, *534*(7605), 115-118.

Fernández-Ruiz, A., Oliva, A., Nagy, G. A., Maurer, A. P., Berényi, A., & Buzsáki, G. (2017). Entorhinal-CA3 Dual-Input Control of Spike Timing in the Hippocampus by Theta-Γ Coupling. *Neuron*, *93*(5), 1213-1226.e1215

Maviel, T., Durkin, T. P., Menzaghi, F., & Bontempi, B. (2004). Sites of neocortical reorganization critical for remote spatial memory [Research Support, Non-U.S. Gov't]. *Science*, *305*(5680), 96-99.

Schuette, P. J., Reis, F. M. C. V., Maesta-Pereira, S., Chakerian, M., Torossian, A., Blair, G. J., Adhikari, A. (2020). Long-Term Characterization of Hippocampal Remapping during Contextual Fear Acquisition and Extinction. *J Neurosci*, *40*(43), 8329-8342.

Stark, E., Roux, L., Eichler, R., Senzai, Y., Royer, S., & Buzsaki, G. (2014). Pyramidal cell-interneuron interactions underlie hippocampal ripple oscillations. *Neuron*, *83*(2), 467-480.